# A Metabolite Perspective on the Involvement of the Gut Microbiota in Type 2 Diabetes

**DOI:** 10.3390/ijms241914991

**Published:** 2023-10-08

**Authors:** Yifeng Fu, Siying Li, Yunhua Xiao, Gang Liu, Jun Fang

**Affiliations:** Hunan Provincial Engineering Research Center of Applied Microbial Resources Development for Livestock and Poultry, College of Bioscience and Biotechnology, Hunan Agricultural University, Changsha 410128, China; fuyifeng@stu.hunau.edu.cn (Y.F.); lisiying@stu.hunau.edu.cn (S.L.); huazipiaoling.123@163.com (Y.X.)

**Keywords:** gut microbiota, dietary diet, metabolites, short-chain fatty acids, bile acids

## Abstract

Type 2 diabetes (T2D) is a commonly diagnosed condition that has been extensively studied. The composition and activity of gut microbes, as well as the metabolites they produce (such as short-chain fatty acids, lipopolysaccharides, trimethylamine N-oxide, and bile acids) can significantly impact diabetes development. Treatment options, including medication, can enhance the gut microbiome and its metabolites, and even reverse intestinal epithelial dysfunction. Both animal and human studies have demonstrated the role of microbiota metabolites in influencing diabetes, as well as their complex chemical interactions with signaling molecules. This article focuses on the importance of microbiota metabolites in type 2 diabetes and provides an overview of various pharmacological and dietary components that can serve as therapeutic tools for reducing the risk of developing diabetes. A deeper understanding of the link between gut microbial metabolites and T2D will enhance our knowledge of the disease and may offer new treatment approaches. Although many animal studies have investigated the palliative and attenuating effects of gut microbial metabolites on T2D, few have established a complete cure. Therefore, conducting more systematic studies in the future is necessary.

## 1. Introduction

This paper discusses the prevalence of type 2 diabetes (T2D) as a widespread metabolic condition [1]. Approximately 500 million individuals globally have been diagnosed with diabetes, and by 2045, that figure is projected to reach 700 million, according to a 2019 estimate. Diabetes, which mainly affects adults, frequently results in high blood glucose levels because of insufficient insulin synthesis and insulin resistance (IR). There are three types of diabetes: type 1, type 2, and gestational diabetes. T2D, which is the most common type, accounts for more than 90% of all diabetes cases and is a serious health issue in industrialized and developing nations [2]. Although genetic susceptibility is a major contributor to T2D, physical inactivity, lifestyle [3], and obesity are also important contributors to T2D [4].

Numerous studies have demonstrated the importance of the gut microbiota as a pathogenic component in the emergence of IR and T2D [5]. The metabolism of carbohydrates, proteins, choline, and major bile acids (BAs) by gut microbes might result in bioactive metabolites that are not digested and absorbed from meals [5]. These metabolites have been suggested to play a crucial role in the development of IR and T2D [6]. The proteolytic fermentation carried out by the gut microbiota leads to the production of various substances, such as indoles, phenols, p-cresols, hydrogen sulfide, branched-chain fatty acids, ammonia, and polyamines [6]. Some of these substances might either help or harm the host’s metabolic and intestinal balance. Gut microbes are essential in regulating host energy homeostasis, appetite, and modulating host immunity through the production of short-chain fatty acids (SCFAs), vitamins, and metabolites (Figure 1) [7]. Metabolic disorders can be influenced by microbial metabolites, which are determined by the composition and arrangement of the gut microbiota [8]. Recent research on microbial metabolites may help us understand T2D better (Figure 2) [2].

The intricate community of microbes that makes up the human intestinal flora is essential to preserving the body’s intestinal homeostasis [9]. It consists of 100 trillion archaea and bacteria, weighing approximately 1.5 kg, making it a microbial organ [10]. There are various cooperative bacteria, archaea, fungi, and viruses throughout the body’s surfaces and cavities, most of which are symbiotic or commensal microorganisms [11]. These microorganisms are similar in size to somatic cells but possess more genes and exhibit diverse metabolic patterns [12]. Numerous research studies have looked at the diversity and makeup of gut bacteria as well as their effects on disease and health, including obesity, inflammation [13], and T2D [14]. The production of small molecule metabolites by gut bacteria profoundly affects the host’s metabolism and immunity. Given the potential of metabolites for disease interventions, it is crucial to explore the genetic potential of gut microbes to produce specific metabolites (Table 1). Doing so will enable a better understanding of the relationship between metabolic diseases and microbiota, leading to the development of novel treatment approaches for metabolic disorders and providing valuable insights into their clinical management.

Gestational diabetes is associated with dysbiosis of the gut microbiota in newborns, according to Naser A. Alsharairi’s study, which also found that the gut microbial metabolite SCFA affects the expression of diabetes-related genes in newborns [15]. Huixia Yang et al. discovered that acetic, propionic, and butyric acids have potential antidiabetic and anti-inflammatory effects in women with gestational diabetes, as demonstrated by their levels in maternal circulation [16]. These SCFAs could be a therapeutic target for treating gestational diabetes. Furthermore, according to a study by Jessica E. Harbison et al., lower levels of circulating SCFAs are also associated with type 1 diabetes (T1D) in youth [17], which emphasizes the potential therapeutic role of gut microbial metabolites, including SCFAs, in different types of diabetes. Xiao et al. found low levels of phenolic acids and SCFAs in patients with T1D. Phenolic acids have demonstrated antidiabetic and anti-inflammatory effects. The reduced production of SCFAs in T1D patients could be attributed to the decreased abundance of SCFA-producing flora resulting from the loss of phenolic acids [18]. Therefore, supplementing with phenolic acid compounds may be an effective approach for treating T1D.

**Table 1 ijms-24-14991-t001:** Examples of research on microbial metabolites associated with metabolic disorders.

Metabolite	Microbial Agent	Effect	References
**Propionate**	*Lactobacillus*	Promotes GLP-1 secretion, improves blood glucose, lipid levels, and intestinal barrier function and increases beneficial bacteria.	[19]
**Butyrate**	*Anaerostipes hadrus*	Reduces glycated haemoglobin levels, improves mood, sleep, and blood sugar levels and increases probiotic abundance.	[20]
**Isovaleric, lactic acids**	*Prevotella copri*	Improves insulin secretion and promotes glucose homeostasis.	[21]
**BA**	*Bacteroides fragilis*	Affects blood sugar levels	[22]
**MAM (** **m** **icrobial anti-inflammatory molecule)**	*Faecalibacterium* *prausnitzii*	Regulation of tight junction protein expression, restoration of intestinal barrier function and resistance to inflammation.	[23]
**Acetate**	*Akkermansia*	Inhibits intestinal inflammation and promotes intestinal epithelial integrity.	[24]
**SCFAs** **and branched-chain** **fatty acids**	*Acidaminococcus* spp., *Clostridia* spp.	Promotes healthy metabolism and amino acid fermentation.	[25,26]
**Tryptamine**	*Ruminococcus gnavus*	Increases bile acid levels and secretion of tryptamine.	[27,28]
**Histamine**	*Morganella morganii*	Decarboxylation of histidine (histidine decarboxylase (HDC))	[29,30]
**Imidazole propionate** **(ImP)**	*Adlercreutziae equolifaciens* *Anaerococcus* *prevotii*	Impairment of glucose tolerance and insulin signalling.	[31]
**Dopamine**	*Enterococcus faecalis*	Regulates glucose uptake, insulin sensitivity and lipid metabolism.	[32,33]

In addition to highlighting the possibility of dietary interventions as a means of preventing and treating T2D, this review critically reviews recent results on the consequences of gut microbial metabolites on the onset and severity of T2D. We found that most of the review literature provided few summary overviews of the relevant studies that have been carried out, so we have provided a summary overview of the relevant studies in Section 5 of the article. The review incorporates information from a wide range of studies on both people and animals to discuss the role of gut microbial metabolites in the prevention of T2D, providing valuable insights for future research. Overall, this study highlights the possibility of dietary treatments as a preventative measure and the significance of comprehending the connection between gut microbiota and metabolic disorders in the development of novel therapeutic approaches for T2D.

## 2. Method

As of September 2023, we conducted a literature search in the Pubmed and Web of Science databases, mostly searching for articles published in English in the last five years, including reviews and research-based papers. The following keywords were searched: “gut microbial metabolites and T2D”, “short-chain fatty acids”, “bile acids”, “branched-chain amino acids”, “intestinal epithelial barrier dysfunction”, to study the effect of gut microbial metabolites on T2D.

## 3. Gut Microbiota in Type 2 Diabetics

Increased hepatic glucose production, increased insulin sensitivity, and inadequate insulin secretion are all symptoms of T2D. The pathophysiology of T2D is multifactorial, involving local and systemic inflammation, glucose metabolism disruption, and other factors. Recent studies have identified dysbiosis as a potential contributor to the emergence of T2D [34]. Thus, understanding the gut microbiota in T2D patients has become an important area of research. Several research investigations have looked at the gut microbiota of T2D patients and found substantial differences between them and controls at the genus level. These findings imply a link between changes in the gut flora and diabetes [35].

The concentrations and types of bacterial populations differ between obese and lean individuals. Those with a reduced abundance of bacteria have considerably greater rates of obesity, IR, and dyslipidemia. A study on oxidative stress in metabolic illnesses highlighted the dysbiosis profile of persons with T2D. This profile is characterized by a decreased butyrate-producing bacteria and an increased opportunistic infection [36]. Metformin treatment leads to many differences in microbial composition. In the untreated subgroup of T2D patients, the *Roseburia*, *Subdoliggranum*, and *Clostridium bolteae* species were enriched, although metformin use was not linked to a decreased prevalence of butyrate-producing bacteria [37]. Chen et al. found that metformin prevented intestinal barrier dysfunction in colitis, attenuated the loss of tight junction proteins, and reduced bacterial translocation and levels of the pro-inflammatory factors IL-6 (interleukin-6), TNF-α (tumor necrosis factor), and IL-1β (interleukin-1β) [38]. Phylogenetic analysis has shown that microbial dysbiosis causes obesity first, followed by IR, and subsequently diabetes. Obesity is a significant contributory factor for T2D [39]. Numerous early classical fecal transplantation studies demonstrate that the gut microbiota is crucial for acquiring energy, accumulating adipose tissue, and reducing IR [40].

Several studies have shown a decline in the ratio of Bacteroides/Firmicutes among the flora compositions connected to metabolic disorders and obesity [41]. For instance, a study involving 277 Danish persons without diabetes found that the gut microbiota of insulin-resistant patients had an increased capacity to manufacture branched-chain amino acids (BCAAs) and elevated serum BCAA levels [42]. On the other hand, many diabetic people have decreased levels of microorganisms that produce SCFAs. SCFAs bind to the G protein-coupled receptor (GPCR), triggering the production of GLP-1, which in turn decreases gluconeogenesis in the liver, increases insulin sensitivity, and provides a central feeling of satiety. Furthermore, SCFAs have been shown to directly block low-level inflammatory responses, and thus, may play a crucial role in preventing inflammation commonly found in metabolic illnesses [43].

Interestingly, a high-fat diet (HFD) leads to a more abundant composition of LPS-containing bacteria, which intensifies the susceptibility to endotoxemia and hypo-inflammation due to IR [44]. Farid Najafi et al. conducted a study that highlights the association between Western diets, which include red meat and processed meats, and the pathogenesis of inflammatory diseases. Their findings demonstrated that individuals with T2D have a significantly higher pro-inflammatory dietary intake than non-T2D individuals; elevated levels of the inflammatory factors IL-1 (Interleukin-1) and TNF-α result from this type of diet, and these factors can interfere with insulin signaling, leading to the development of IR [45]. Demirer et al. conducted a study that investigated the effects of dietary advanced glycation end products (AGEs) on inflammation. Their findings suggest that the intake of dietary AGEs can directly or indirectly induce inflammation; given that individuals with diabetes are prone to oxidative stress and inflammation, excessive consumption of dietary AGEs may expedite the inflammatory process [46]. Based on these findings, we can conclude that those with T2D have a significantly different gut microbiota makeup from those who are healthy.

## 4. Epithelial Barrier Dysfunction in Diabetic Patients

The intestinal mucus barrier, the intestinal vascular barrier, and the intestinal epithelial barrier make up the intestinal wall, which separates the intestinal contents from extraintestinal tissues and organs. Intestinal mucus, which is produced by goblet cells and is the main element of the intestinal barrier, is essential for defending the gut wall from infections. The thick inner and outer mucus layers of the colon are where intestinal microorganisms settle. These layers contain different anti-microbial peptides (AMP) and secreted IgA (immunoglobulin A), which are chemically immune to assault by pathogenic bacteria, providing an additional defense against harmful microorganisms [47]. Under typical circumstances, the first intestinal barrier, which is composed of intestinal mucus, provides a physical and chemical defense against pathogens. In addition to the mucus barrier, the second intestinal barrier is made up of sterile epithelial cells and is sealed by junctional complexes, such as adherent junctions, gap junctions, tight junctions (TJ), and bridge particles. The permeability of cells and the passivity of nutrients, ions, and water are regulated by the intestinal epithelial membrane, supported by the junctional complexes. Meanwhile, immune cells including macrophages, dendritic cells, and lymphocytes present in the gut’s epithelial lamina propria release different antimicrobial peptides, IgG (immunoglobulin G), and cytokines to actively participate in maintaining intestinal immunological homeostasis [48].

With T2D, a combination of internal and external causes results in damage to the intestinal barrier. Diet is the most direct influence, as a Western-style diet not only tends to induce obesity but also leads to an imbalance of intestinal bacteria and increased intestinal permeability [49]. The type of dietary fat can also affect the level of endotoxemia. For example, mice consuming a diet high in saturated fatty acids, such as lard, experienced an increase in endotoxemia compared to mice with a diet high in polyunsaturated fatty acids, such as fish oil [50]. In addition to lipids, dietary fiber is an essential regulator of intestinal integrity [51]. Supplementation of the diet with oligofructose has been shown to lower intestinal permeability and the plasma levels of LPS. The integrity of TJ and adherent junctions is inhibited by organismal hyperglycemia in a way that is dependent on the bidirectional glucose transporter protein (GLUT2) in T2D [52]. Non-alcoholic fatty liver disease (NAFLD) caused by a HFD is seen in animals with bacterial intestinal abnormalities and intestinal bacteria found in extraintestinal organs. However, the introduction of the fecal microbiota from healthy mice into sick animals has been shown to repair the damage to the intestinal barrier [53].

Due to the breakdown of the intestinal barrier, which allows intestinal bacteria to enter the bloodstream, T2D and inflammatory diseases are thought to be primarily caused by the bacteria in the intestines and their pathogen-associated molecular patterns (PAMPs), which include LPS. In vitro and in vivo investigations on mice showed a relationship between increased intestinal LPS and intestinal TJ expression [43]. Therefore, it is likely that the intestinal vascular barrier allows the passage of pathogens, pro-inflammatory cytokines, and toxic metabolites into the bloodstream. The presence of LPS on bacterial surfaces stimulates macrophages, leading to the release of inflammatory cytokines such as IL-6, TNF-α, and IL-1. Because of the activity of macrophages, dendritic cells, and other cells associated with inflammation, this in turn leads to an inflammatory milieu in metabolic tissues that worsens, as shown in Figure 3 [54].

Studies have also shown that infusing beneficial bacteria in vitro can produce specific metabolites that improve IR and hyperglycemia. Various molecular mechanisms of probiotic intervention in T2D have been explored, including the restoration of the intestinal barrier, prevention of the inflammatory response, reduction in oxidative stress, and production of advantageous metabolites. These results highlight the significance of gut bacterial imbalance as a cause of intestinal barrier disturbance (Figure 4) [5].

Hu et al. showed that a high-sugar diet induced symptoms of T2D in rats, causing inflammation and disturbances in glucose metabolism and lipid metabolism, as well as a higher abundance of harmful bacteria present in the gut [55]. Guo et al. found that a methionine/choline deficient diet resulted in the expression of pro-inflammatory factors in mice, while a methionine/choline deficient diet resulted in elevated serum insulin levels in mice with symptoms similar to those of T2D [56]. Estaphan et al. found that a diet high in iron caused insulin damage and inflammatory cell infiltration in rats, increasing the risk of developing T2D [57]. In conclusion, the disruption of the intestinal barrier can be attributed to intestinal bacterial dysbiosis caused by a high-sugar and HFD. This dysbiosis further leads to the disruption of intestinal permeability and the tight junction structures. However, it is important to note that this dysfunction in the intestinal epithelium can be reversed through appropriate dietary treatment and flora transplantation.

## 5. Effect of Gut Microbial Metabolites on Type 2 Diabetes

### 5.1. Short-Chain Fatty Acids

The most important SCFAs, including acetate, propionate, and butyrate, are generated by gut bacteria from indigestible food. These SCFAs are among the most abundant metabolites produced by gut microorganisms. The health effects of SCFAs have been widely reported. These tiny molecule substances control several metabolic processes in the gut and many other organs, including the liver, adipose tissue, muscle, and brain (Figure 5) [58]. Many research articles have discussed the various roles of SCFAs in IR and T2D. These roles include regulating pancreatic β-cell proliferation and insulin secretion, managing immunomodulatory processes, and preserving the intestinal epithelial integrity [58].

SCFAs bind to the GPCR’s free fatty acid receptor 3 (FFAR3) and GPR43 (free fatty acid receptor 2 or FFAR2), which are expressed in a variety of cell types, such as intestinal epithelial cells and enteroendocrine cells [59]. Through GPR41, SCFAs can produce a substance called leptin, which regulates food intake and energy expenditure [60]. A mouse study showed that animals without GPR43, under normal dietary circumstances, became obese, whereas mice with above-normal expression of GPR43 remained slim even when consuming fatty diets [61]. This study demonstrates that the activation of GPR43 by SCFAs inhibits specific signals from insulin in adipose tissue, thereby preventing excessive fat accumulation. Additionally, it significantly enhances the efficiency of energy use in other tissues, which is beneficial for maintaining metabolic homeostasis in the body.

Similar to this, it has been demonstrated that activating GPR41 significantly affects a number of physiological processes, including gluconeogenesis, energy expenditure, and the production of the intestinal PYY [62]. Additionally, several studies have demonstrated that SCFAs can enhance glucose metabolism in adipose, muscle, and liver tissues, thus promoting body homeostasis [63]. Moreover, SCFAs have anti-inflammatory properties and can reduce mucosal and chronic organismal inflammation by inhibiting pro-inflammatory cytokines and promoting the secretion of anti-inflammatory cytokines [64]. These properties make SCFAs a promising therapeutic agent for inflammatory diseases, including diabetes. In a study conducted on T2D patients, a long-term colonic infusion of propionate was found to be beneficial in reducing body weight in overweight individuals and minimizing the detrimental consequences of IR [65]. Hence, targeting the GPR41 receptor with SCFAs may have significant therapeutic benefits for individuals with T2D.

Chen et al. found that persistent chronic low-grade inflammation in individuals with T2D plays a crucial role in the development of the disease. In T2D, there are elevated levels of inflammatory factors such as IL-6, IL-1β, and TNF-α. The researchers observed that treatment with probiotics resulted in a reduction in these inflammatory factors. Additionally, they discovered that the levels of SCFAs also impact glucose metabolism and IR [66]. However, certain SCFAs, like propionate and butyrate, have been found to have anti-inflammatory actions by reducing the expression of TNF-α, IL-1β, and IL-6 [67]. Additionally, studies have shown that agonists of the GPR119 receptor can increase pancreatic β-cell function and insulin production by promoting the release of intestinal GLP-1, leading to a reduction in hyperglycemia [68]. Therefore, through triggering GPCRs, SCFAs can improve insulin sensitivity, stop the growth of white adipose tissue, and lower inflammation, all of which lead to an improvement in T2D patients’ glucose metabolism.

The circulating levels of SCFAs were significantly lower in patients with T2D, where propionic acid levels showed a trend of negative correlation with IR [69]. In addition, a clinical study discovered a link between stool SCFA levels, body mass index, and IR [70]. However, the relevance of SCFAs in IR and T2D is still being debated and deserves further research. SCFAs regulate the body’s calorie-burning mechanism by binding to GPCRs and initiating specific metabolic pathways. Furthermore, they influence metabolic inflammation by modulating the expression of cytokines that are pro-inflammatory [71].

### 5.2. Lipopolysaccharide

LPS, often known as endotoxin, is a structural component that makes up Gram-negative bacteria’s outermost membrane. It possesses a strong affinity for several immune-related receptors, including Toll-like receptors (TLRs), which are abundant in macrophages and dendritic cells, as well as intracellular pattern recognition receptors called NOD-like receptors and NLRP3 (NOD-like receptor family pyrin domain containing 3) inflammasomes. An in vitro molecular investigation has revealed that LPS reduces insulin sensitivity by activating the TLRs. LPS binds to LPS-binding proteins on macrophages and transmits stimulatory signals through TLR4, located in the lipid bilayer of the cell membrane. This binding initiates an immune response within macrophages. Furthermore, TLR4 activation can induce the production of reactive oxygen species (ROS) and nitric oxide (NO), which contributes to the elimination of invading pathogens and the regulation of immune responses [72].

Toxins and germs cannot enter the bloodstream due to tight junction proteins controlling intestinal permeability in the intestinal epithelium. It was found that patients with T2D had significantly higher levels of LPS, increased levels of zona occludens 1 (ZO-1), a marker of intestinal permeability, and upregulated levels of the inflammatory factor TNF-α, compared to subjects with normal glucose tolerance [73]. Multiple studies have reached the same conclusion, showing that T2D patients and mice with compromised gut barriers have an imbalance of LPS in the peripheral circulation, which is caused by excessive amounts of LPS compared to the microbiota of the gut. As a result, LPS enter the injured gut and cause IR in the pancreas, which aids in the progression of T2D [74].

In a subsequent investigation, it was discovered that T2D patients had higher postprandial LPS levels than typical healthy participants [75]. Campa et al. conducted a study on acutely inflamed mice treated with a HFD and LPS; the researchers observed that these mice exhibited several adverse effects, including increased body weight, impaired insulin sensitivity, acute endotoxemia, upregulated expression of the inflammatory marker TLR-4, and impaired glucose homeostasis [76]. These findings indicate that diabetes causes an increase in intestinal permeability, allowing bacteria and LPS to easily enter the bloodstream. This, in turn, raises serum LPS levels and impairs insulin signaling and glucose metabolism. In conclusion, managing intestinal permeability may have therapeutic potential in reducing the onset and progression of T2D.

### 5.3. Bile Acids

The gallbladder functions as a storage organ for BAs, which are steroid compounds formed by cholesterol in the liver cells. Only bacteria that can adapt to a high concentration of BA in the intestine, due to its inhibitory action on intestinal bacteria, can survive [77]. The gut microorganisms convert primary BAs into secondary BAs. The human body generates primary BAs, such as CA (cholic acid) and CDCA (chenodeoxycholic acid), through two processes: the traditional pathway (by cytochrome P450 family 7 superfamilies A polypeptide 1 or CYP7A1 (Cholesterol-7α-hydroxylase)) and the alternative pathway (through CYP27A1 (Sterol 27-hydroxylase)). Coupling primary BAs with glycine or taurine leads to the formation of TCDCA (tauro-chenodeoxycholic acid), taurocholic acid (TCA), glycocholic acid (GCA), and GCDCA (glycochenodeoxycholic acid). Bile salt hydrolases (BSH), intestinal bacteria that convert BAs into free BAs and secondary BAs, produce BAs such as deoxycholic acid (DCA), lithocholic acid (LCA), and ursodeoxycholic acid (UDCA) from certain BAs that enter the small intestine [78].

More than 90% of the BAs in the gut are absorbed through the distal ileum’s apical sodium-dependent BA transporter (ASBT) [79]. BA ligands, such as FXR (farnesoid X receptor), VDR (vitamin D receptor), PXR (pregnane X receptor), TGR5 (G protein-coupled receptor 5), and S1PR (G protein-coupled S1P receptor), can bind to nuclear receptors and cell surface receptors. The interaction between BA, FXR, and TGR5 directly regulates glucose homeostasis in the intestine, liver, and pancreas. Specifically, when secondary BAs bind to FXR and TGR5, intestinal L cells are stimulated to release the enteric insulinotropic hormone GLP-1, which improves body glucose homeostasis [80].

FXR-deficient animals have slower glucose uptake kinetics in the gut. Mice deficient in FXR have slowed glucose uptake due to the inhibition of intrahepatic glycolysis and reduced glucose uptake and utilization [81]. Additionally, FXR decreases postprandial glucose uptake and inhibits hepatic glycolysis, while FGF15/19 (fibroblast growth factor-15/19) boosts gluconeogenesis [81]. In response to glucose, the pancreatic beta cells’ FXR and TGR5 increase the production of glucagon and insulin [82]. Furthermore, LCA increases VDR by increasing CYP3A synthesis and activating the ERK 1/2 pathway, which obstructs the insulin signaling pathway. These findings suggest that secondary BAs play a crucial role in regulating body glucose homeostasis and reducing the incidence of T2D, through their interactions with receptors such as FXR and TGR5. The activation of these receptors by secondary BAs modulates various metabolic processes involved in glucose regulation. Together, these effects contribute to the maintenance of glucose homeostasis and provide potential therapeutic targets for T2D treatment.

### 5.4. Branched-Chain Amino Acid (BCAAs)

The BCAAs, which include leucine, isoleucine, and valine, are essential amino acids necessary for protein synthesis [8]. However, increased amounts of circulating BCAAs have been connected to metabolic disorders such T2D and IR [83]. The dysbiosis of the gut microbiota, brought on by IR, can significantly increase the concentration of harmful compounds in the bloodstream, including BCAAs. 3-Hydroxyisobutyrate is an intermediate in the catabolism of BCAAs, and in diabetic subjects, increased plasma 3-Hydroxyisobutyrate levels may reflect increased protein catabolism due to relative insulin deficiency resulting from IR-associated insulin secretion deficiency [83]. Inactivation of enzymes involved in BCAAs oxidation was also observed in T2D patients and animal models [84]. In addition, it has been found that an improved peripheral insulin sensitivity in T2D patients is associated with a reduction in circulating BCAAs in the body [85].

T2D individuals’ changes in BCAA levels may be caused by a decreased inhibition of proteolysis, due to IR or impaired lipocalin signaling leading to reduced peripheral tissue BCAA catabolism. Wu et al. found that plasma concentrations of BCAAs affect endothelial function in animals; the researchers reduced plasma levels of BCAAs and enhanced endothelial cell nitric oxide synthesis and insulin sensitivity in rats by adding AKG (a substrate for BCAA transaminase) to their drinking water [86]. Clinical investigations have shown an association between elevated BCAA levels and IR [1]. Zhai et al. found that limiting the concentration of BCAAs in HFDs maintained blood glucose and insulin at stable levels in mice and prevented HFD-induced obesity, lipid inflammation, and IR [87]. Even though the mice’s daily food continued to be high in fat and sugar, the consumption of these amino acids was reduced. The obese mice’s metabolic health was therefore restored, and their glucose tolerance and responsiveness to insulin were both dramatically improved. The genetic risk score (GRS) was not directly connected with the homeostatic model of IR, according to the findings of a genetic risk assessment for circulating BCAA levels and IR. However, a substantial association between higher plasma BCAA levels and the GRS for IR characteristics was discovered [88]. This suggests that while there is no direct causal relationship between higher levels of BCAAs and IR, high levels of BCAA circulation are indirectly impacted by IR.

### 5.5. Trimethylamine N-Oxide (TMAO)

Only gut bacteria have the ability to produce TMA, which is derived from dietary choline, phosphatidylcholine, and carnitine. Once generated by the gut microbiota, TMA is converted into TMAO through the liver’s flavin-containing monooxygenase 3 (FMO3) [89]. The breakdown of TMAO is essential for maintaining glucose homeostasis in the host, a process facilitated by FMO3 [90]. A study conducted on mice demonstrated that a deficiency of FMO3 protected them from obesity and IR [91]. Palika et al. found that patients with T2D with chronic kidney disease had higher serum levels of TMAO, and levels of the inflammatory factors IL-6 and TNF-α were significantly and positively correlated with TMAO levels [92]. Dietary modifications that lead to a decrease in TMAO have been found to improve insulin sensitivity in individuals with T2D. TMAO was linked to oxidative stress, IR, and glucose tolerance, according to a prospective cohort research study [93]. Silencing FMO3 in mice, as shown by Alina et al., resulted in reduced levels of circulating TMAO, glucose, and insulin [90]. According to epidemiological studies, higher levels of TMAO in the body adversely affect T2D [93]. Shan et al. found that in rats with experimentally induced T2D, the relative abundance of *Clostridiales* and *Desulfovibrionales*, colonies associated with TMAO production, was increased and plasma TMAO levels were elevated in the diabetic group compared with control rats, and these promoted atrial inflammation [94]. Elevated serum TMAO levels in T2D patients are strongly associated with cardiovascular risk [95]. In a study of T2D subjects, Adriana et al. found that T2D patients had higher levels of TMAO and an increased risk of NAFLD was observed in T2D patients with elevated levels of TMAO. Elevated levels of TMAO also affected the body’s metabolism of circulating BAs [96].

In summary, elevated levels of TMAO have been found to impact the development of T2D to a certain extent. However, the precise mechanism underlying the influence of TMAO on T2D remains unknown, necessitating further exploratory studies in this area.

## 6. The Influence of Gut Microbes on the Progression of Diabetes

### 6.1. Animal Studies

Gut microbial metabolites are associated with disorders of glucose metabolism [5]. One such study conducted by Lin et al. reported that modifying BAs using Poria oligosaccharides improved IR, glucose intolerance, and IR in HFD mice [97]. Similarly, Wang et al. found that both berberine and metformin reduced food intake, blood sugar levels, and body weight in db/db mice [24]. Moreover, these medications effectively raised LPS levels in the blood, reduced intestinal inflammation, rebuilt intestinal barrier components, and recovered intestinal SCFA levels. In addition, it has been found that SCFAs can be influenced by certain herbal ingredients, such as resveratrol and reishi. Moreover, these herbal ingredients can improve the metabolic disorders associated with diabetes by affecting the metabolites produced by the intestinal microbiota. For example, a study revealed that T2D rats who received a dietary intervention that included beans rich in gamma-aminobutyric acid (GABA) experienced significant benefits. The rats did not gain significant weight, showed reduced levels of total serum cholesterol and fasting blood glucose, and maintained consistent blood sugar levels. These beneficial effects were achieved through alterations in the intestinal microbiota [98,99]. These findings suggest that modulating microorganisms and their metabolites can have a positive impact on diabetic complications.

Several studies have investigated how microbial metabolites can affect glucose metabolism. In mice without FFAR3 receptors, butyrate and propionate have been found to boost the release of intestinal hormones while also reducing hunger, diet-induced weight gain, and glucose intolerance [100]. Additionally, a study on bifidobacterial intervention in mice illustrated that increasing acetate concentration in the cecum and plasma can lead to a reduction in fat levels. Furthermore, propionate has been shown to impact the development of diabetes by increasing the expression of gluconeogenic genes and AKT phosphorylation in response to insulin stimulation in HepG2 hepatocytes through the GPCR43-mediated AMPK signaling pathway [14]. Indolepropionic acid, a class of tryptophan-derived metabolites, has also been investigated for its effects on glucose metabolism. It was discovered to lower insulin and blood glucose levels during fasting in one rat investigation, indicating that it might have an immediate effect on the production or absorption of glucose [101]. Additionally, in another study with streptozotocin-induced diabetic rats, two weeks of indole propionate supplementation led to decreased oxidative damage, reduced endoplasmic reticulum stress and apoptotic markers, and improved mitochondrial function. It also changed the pain behavior of the diseased group of mice [101].

Investigations have shown that TMAO increases with increasing levels of IR in humans and animals [102,103]. Treatment with TMAO promoted glucose intolerance in mice, while reducing TMAO prevented glucose intolerance. Furthermore, phenylacetic acid, a product of microbial metabolism and an aromatic compound produced from Bacteroides, has been found to have negative effects on glucose metabolism. In an animal study, phenylacetic acid increased triacylglycerol levels in liver tissue and reduced insulin-induced AKT phosphorylation. In a study conducted by Zhang et al., db/db mice displayed considerable liver lobule hypertrophy and steatosis, as well as significant reductions in glycogen reserves and a significantly lower GPR43 expression. However, treatment with sodium butyrate significantly increased glycogen stores and the levels of the butyrate receptor, the GPR43, in mice and HepG2 cells. Additionally, it was discovered that sodium butyrate increased the expression of the sodium–glucose co-transporter protein 1 (SGLT1) and GLUT2 on cell membranes [104], which play important roles in glucose transport. By encouraging glycogen metabolism in hepatocytes, the bacterial metabolite sodium butyrate has been found to help maintain glucose levels in balance [104].

Probiotics, prebiotics, and medicines are useful therapies for T2D, according to numerous research studies [100]. Wu et al. revealed that therapy with the ethanolic extract of *Clostridium perfringens* decreased the levels of BCAAs in the intestinal contents of HFD-induced diabetic mice. As a result, the treatment had a therapeutic effect by alleviating the disruption of glucolipid metabolism [105]. In a related study, Stenman et al. discovered that giving mice probiotics and metformin enhanced their mice’s insulin sensitivity and glycemic management. They suggested that the therapeutic effect might be related to the secretion of intestinal proinsulin [106]. The study found that fermenting camel milk with probiotics affected diabetes by inducing GLP1 production among streptozotocin-induced rats [19]. Therefore, the modulation of metabolites produced by gut microbes through probiotics as well as drugs can be an effective means of treating T2D [107]. It is clear from this that gut microbial metabolites serve a purpose for treating T2D, and numerous research has shown their therapeutic advantages, as shown in Table 2.

### 6.2. Human Studies

Several gut bacteria, including those that affect insulin sensitivity, have been demonstrated to have anti-diabetic effects in people through variety of pathways [108]. In human research, it has been shown that butyrate and propionate are crucial for glucose metabolism. In an animal study, dietary fiber was found to improve lipid metabolism by promoting SCFA production [109]. The composition of the gut microbiota, which is crucial for the development and maintenance of a healthy intestinal flora, is significantly influenced by food. Moreover, dietary choices can directly impact on the functionality and diversity of the gut microbiota. Studies have shown that dietary fiber obtained from natural vegetables can have an impact on the survival and metabolism of probiotics in the gut and can increase the production of SCFAs, thus benefiting the organism [110].

One mechanism through which dietary changes affect the gut microbiota is by providing the necessary substrate for fermentation. Dietary fiber serves as a substrate for the complex reactions undergone by microorganisms during the fermentation process. On the other hand, sugars found in foods like fruits and whole grains, which are difficult for the body to digest and absorb, are metabolized by the gut microbiota in the stomach. SCFAs, a molecule that is produced as a result of this metabolism, are essential for maintaining the healthy operation of the intestinal membrane and reducing inflammation [111]. The intake of SCFAs effectively produces GLP1 and PYY, which enhance satiety through the gut-brain axis and subsequently reduce hunger and food intake. Increased plasma levels of BCAAs have been associated with higher risks of T2D and IR in some human examinations [112,113]; therefore, reducing BCAA intake can restore metabolic health and improve glucose tolerance and insulin sensitivity. Additionally, dietary TMAO interferes with insulin-related signaling pathways in the liver, raising fasting insulin levels and the level of peripheral IR, which can cause inflammation in adipose tissue [114].

The etiology of the metabolic syndrome is influenced by various factors, including indoles and their derivatives. The association between *Bacillus butyricus* and the remission of IR was particularly striking [115]. Additionally, in a study of diabetic individuals from Finland, indolepropionic acid was found to have a protective effect on the development of T2D. This protective effect may be due to indolepropionic acid’s ability to control enteroendocrine L-cell production, which preserves insulin secretion [116]. In conclusion, targeted dietary modifications and specific metabolite synthesis by gut bacteria can significantly contribute to the management of diabetes.

In addition to the measure of regulating gut microbial metabolites, increased physical activity can help reduce the incidence of diabetes. Inactivity, overweight, and an inadequate diet can lead to chronic oxidative stress. This stress can impact insulin secretion and the action of insulin on receptor cells, increasing the likelihood of macrovascular and microvascular complications [117]. Since obesity increases the chance of developing IR, and a number of metabolic diseases that affect glucose metabolism, exercise can also help people lose weight [118]. Furthermore, in persons with T2D, short-term aerobic exercise training enhances mitochondrial function and increases insulin sensitivity. For example, seven days of intense aerobic exercise can lower blood sugar levels without causing weight loss. This is because it increases the amount of insulin-stimulated glucose handled and reduces the amount of hepatic glucose produced [119].

In conclusion, influencing gut microbial metabolites by means of exogenous diet or interventions can affect the development of T2D, as shown in Table 3.

## 7. Conclusions

Dysbiosis of the microbiota in the intestines, which alters metabolite synthesis, is a substantial contributor to the onset of T2D. Both dietary changes and medication that alters the microbiome of the gut can prevent and manage T2D. According to studies, intestinal barrier failure in diabetics can be repaired by complex chemical processes involving gut microbial metabolites, which have been found to play a major part in the development of diabetes. In addition to clinical hypoglycemic drugs and herbal ingredients, a diet rich in dietary fiber can significantly prevent diabetes. It is noteworthy that appropriate physical activity and a change in unhealthy lifestyle habits are also effective measures for preventing T2D. Patients with T2D have abnormal insulin secretion, disturbed glucose metabolism, and develop IR. Gut microbial metabolites, namely SCFAs, can modulate low-level inflammatory responses and improve glucose metabolism. BAs have the ability to interact with receptors such as FXR and VDR, allowing them to regulate glucose homeostasis in both the gut and liver. On the other hand, high levels of LPS can lead to increased intestinal permeability, resulting in bacteremia and impairing insulin signaling pathways. Furthermore, abnormal levels of BCAAs have been associated with an increased risk of metabolic disorders in the organism. Additionally, elevated levels of TMAO have been linked to the development of IR. To effectively treat T2D, it is crucial to maximize the beneficial role of metabolites like BAs and SCFAs, while simultaneously reducing the levels of potentially harmful metabolites, namely LPS, BCAAs, and TMAO.

Therefore, we hypothesize that this can provide a thought for future researchers. Since gut microbial metabolites have an important role in T2D, there is a need to develop a targeted therapeutic agent based on gut microbial metabolites in the future. In addition, because of the large number of gut microbial species, the exact mechanism of which specific gut microbes treat T2D is unknown, and the question of whether there are other undiscovered metabolites that may have an impact on T2D remains to be examined.

## Figures and Tables

**Figure 1 ijms-24-14991-f001:**
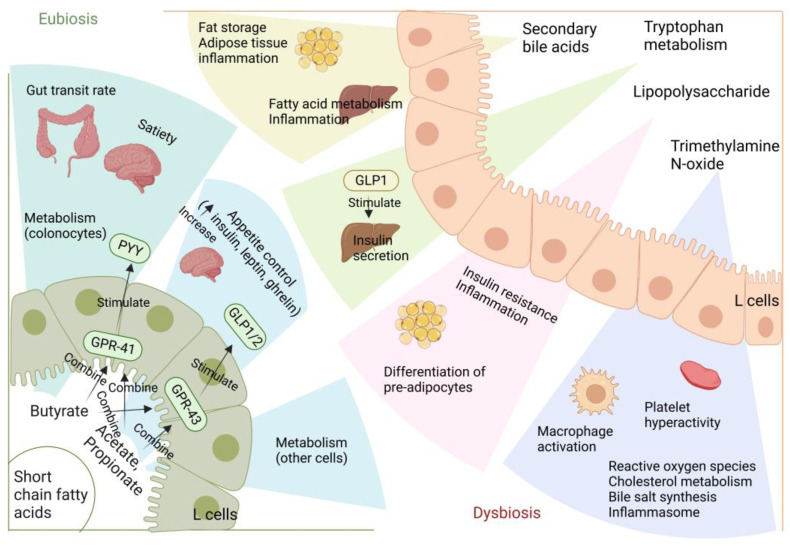
Metabolites of the gut microbiota in dysbiosis and eubiosis. GPR: G protein-coupled receptor; GLP: glucagon-like peptide; PYY: peptide YY.

**Figure 2 ijms-24-14991-f002:**
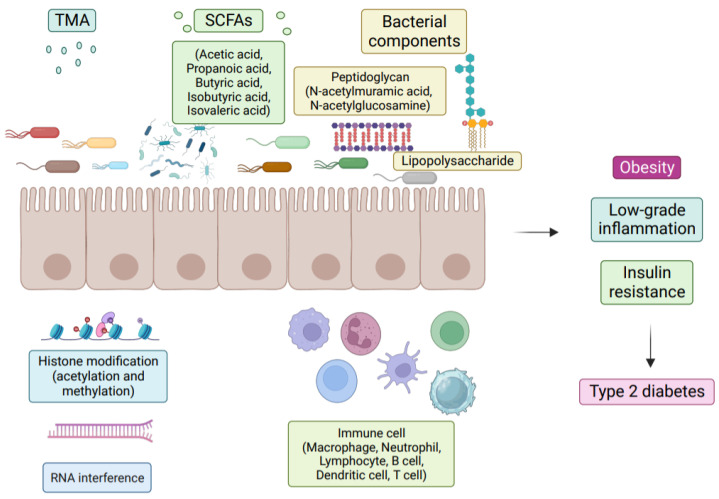
Effects of gut microbiota, microbial metabolites, and bacterial components on T2D. The host enzyme specifically converts the microbial metabolite TMA (Trimethylamine) to TMAO (Trimethylamine N-Oxide), and higher TMAO levels have been seen in people with IR. On the other hand, some bacterial metabolites such as SCFAs have been found to improve glucose homeostasis and IR by influencing epigenetic programming. This is due to their ability to inhibit the activity of the histone deacetylase enzyme. Furthermore, it is not just live bacteria that can affect health outcomes. Bacterial substances such as LPS (lipopolysaccharide), flagellin, and peptidoglycan have been identified as potential causes of an inflammatory reaction, thereby increasing the risk of developing T2D. In conclusion, the role of gut microbiota and its various components in health and disease is complex and requires further study to better understand its mechanisms and potential therapeutic implications.

**Figure 3 ijms-24-14991-f003:**
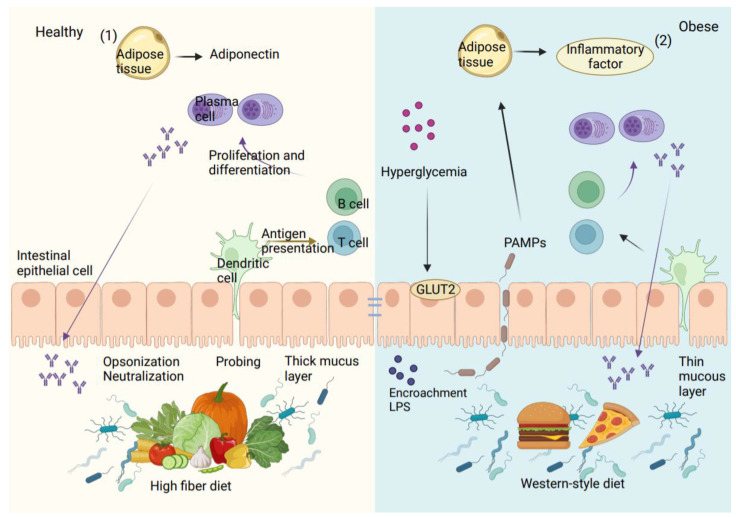
The intestinal barrier is disturbed in people with obesity and diabetes. (**1**) A high-fiber diet can support the function of the intestinal barrier by enhancing the expression of tight junction proteins and immune cell function. Antigen-presenting cells like dendritic cells constantly monitor the intestinal environment, presenting antigens to T and B cells, which can lead to immune tolerance or inflammation based on cytokine and antibody expression. To promote intestinal health, consuming foods high in fiber can be beneficial. (**2**) Thinner mucus layers in the intestines of those who have metabolic syndrome make it possible for opportunistic microorganisms to invade the intestinal lining and cause infections. Additionally, levels of IgA-positive B cells and IgA secretion are lower, leading to microbial alterations and the outgrowth of opportunistic pathogens. A Westernized diet can also cause the intestinal lining’s tight junction proteins to express less frequently, which can cause bacteria and PAMPs to move about. In diabetics, hyperglycemia (high blood glucose levels) can also increase bacterial translocation by decreasing tight junction expression via GLUT2. PAMPs can cause inflammation in various tissues, including adipose tissue, where macrophages multiply and amass. Adipose tissue macrophages in particular are in charge of low-grade inflammation, which is marked by elevated levels of pro-inflammatory cytokines and decreased levels of anti-inflammatory cytokines.

**Figure 4 ijms-24-14991-f004:**
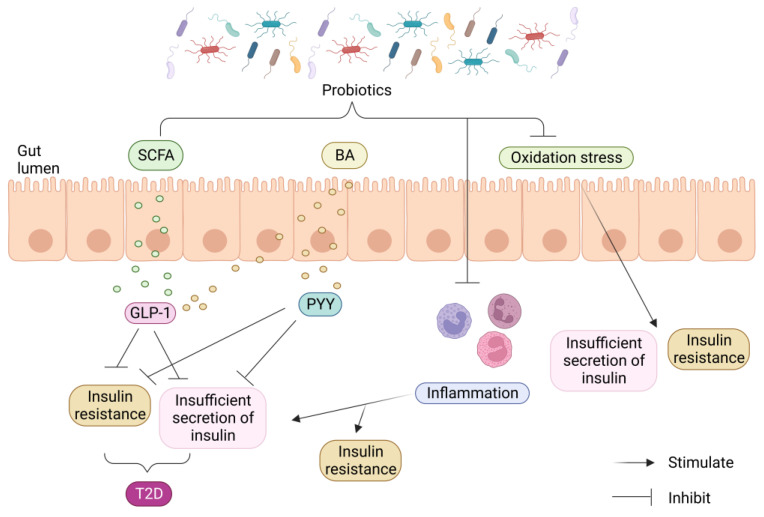
Probiotics’ impact on T2D is mediated at the molecular level. Probiotics help the gut produce healthy metabolites including SCFAs and certain BAs, which boost the release of GLP-1 and PYY and lessen IR and dysfunctional insulin secretion. Probiotics also lessen generalized inflammation by changing the bacteria in the stomach.

**Figure 5 ijms-24-14991-f005:**
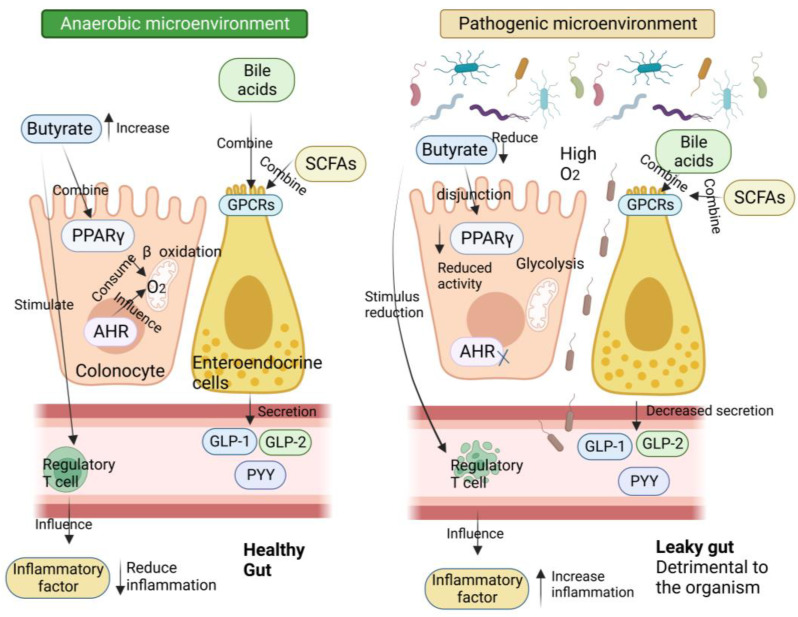
Molecular mechanisms linking gut microbiota and host health in both healthy and pathological situations. In healthy situations, colonocytes use butyrate as an energy substrate through beta-oxidation in the mitochondria. This process consumes oxygen and helps maintain an anaerobic condition in the lumen. Additionally, butyrate binds to peroxisome proliferator-activated receptor gamma (PPARγ), which represses inducible nitric oxide synthase (iNOS), leading to a decrease in nitric oxide (NO) and nitrate production. Glycolysis increases, oxygen consumption lowers, and PPARγ activity decreases in pathological conditions with low butyrate levels in the lumen. Consequently, iNOS expression increases, leading to more NO production and an increased availability of nitrates for specific pathogens. Furthermore, butyrate can stimulate regulatory T cells (Treg) to reduce inflammation. In healthy colonocytes, the nuclear transcription factor aryl hydrocarbon receptor (AhR) is extensively expressed and active. The function of the intestinal barrier can change as a result of decreased AhR activity or agonist deficit. SCFAs, particularly endocannabinoids (eCBs), and BAs all activate a number of important receptors that are expressed by enteroendocrine cells (L-cells). The activation of these receptors increases the secretion of important gut peptides like GLP-1, GLP-2, and PYY. Together, the interactions between these molecular players and gut bacteria help to prevent metabolic endotoxemia and hepatic steatosis, decrease intestinal permeability, and increase insulin secretion and sensitivity. They also help people eat less and have lower plasma lipid levels. There is a correlation between all of these effects and less inflammation. On the other hand, opposing effects have been seen under pathological circumstances.

**Table 2 ijms-24-14991-t002:** Examples of animal studies.

Exogenous Substances Affecting Metabolites	Regulated Substance	Symptoms of Improvement	References
Poria oligosaccharides	BAs	Improves glucose intolerance and IR, lowers blood glucose levels, reduces intestinal damage and restores normal microbiota balance.	[97]
Berberine and metformin	LPS	Reduces food intake, body weight, blood glucose levels, restores SCFA levels, reduces intestinal inflammation and restores intestinal barrier function.	[24]
Legumes diet	Gut microorganisms	Reduces body weight, lowers triglyceride and total cholesterol levels, increases probiotic abundance and regulates glucolipid metabolism.	[98,99]
Sodium butyrate	Glycogen	Upregulation of GPR43 and GLUT2 expression promotes hepatocyte glycogen metabolism and maintenance of blood glucose homeostasis.	[104]
Extract of *Sargarsum fusiforme*	BCAAs	Reducing food and water intake lowers fasting blood glucose levels while improving glucose tolerance and hepatic oxidative stress.	[105]
Probiotics	Intestinal proinsulin	Improves blood glucose and lipid levels, enhances insulin secretion, upregulates claudin-1 and mucin-2 expression, and improves intestinal barrier function.	[19]
Probiotics and prebiotics	Gut microbial metabolites	Reduces intestinal endotoxin concentrations, energy gain, and pro-inflammatory factor levels while increasing insulin sensitivity.	[100]

**Table 3 ijms-24-14991-t003:** Examples of human studies.

Exogenous Substances Affecting Metabolites	Regulated Metabolites	Impact on T2D	References
Dietary fiber	SCFAs	Improve the growth, survival, and metabolic environment of probiotics and promote sugar metabolism.	[110]
Fruits and grains	SCFAs	Reduced food intake, lower levels of inflammation and maintenance of intestinal mucosal homeostasis.	[111]
Excessive intake of protein	BCAAs	Affects glucose tolerance, interferes with metabolism and increases the risk of IR.	[112,113]
Red meat foods	TMAO	Increases the risk of T2D, interferes with hepatic insulin signalling pathways, induces adipose tissue inflammation and impairs glucose tolerance.	[114]
Diet	Indolepropionic acid	Maintains normal β cell function, reduces risk of T2D and lowers inflammation levels.	[116]

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
