# Peer review of "A Metabolite Perspective on the Involvement of the Gut Microbiota in Type 2 Diabetes"

_ijms, 2023, doi:10.3390/ijms241914991_

Round 1
Reviewer 1 Report
A. Summary
The manuscript on hand is a narrative review presenting the relevance of microbiome-derived bacterial metabolites in type 2 diabetes. Both animal and human studies and potential intervention approaches aimed at modulating the microbiota and its metabolite production are discussed. The general topic of the review is of general interest, as type 2 diabetes affects a significant number of the populations worldwide, with expected further increasing prevalence. Understanding the pathophysiological mechanisms underlying disease, including those associated with the gut microbiome, is important in the context of therapeutic approaches.
B. General concept comments
The aspects recounted here have already been extensively covered in a large number of other reviews and therefore provide little additional understanding beyond previous work. The corresponding original works are often not cited (citation of reviews) or incorrect publications are referenced. In some cases, references are completely missing. In terms of language, the manuscript is at times not appropriate for a scientific publication and warrants major language editing and restructuring of content. It is suggested that the authors address a more specific aspect that has not been addressed as frequently. In doing so, the authors should primarily examine the relevant original work in order to become better acquainted with the topic.
C. Specific Comments
Since the overall content should be reconsidered and revised, the following are only exemplary more specific comments on the content.
The authors' conclusions are often too strong and cannot be inferred based on the studies that are provided.
- Line 314: “Secondary bile acids […] are responsible for the onset of T2D”.
- Line 451: “Therefore, reducing the number of microbial metabolites generated in the gut […] may be a promising approach for treating T2D”.
- Line 385 ff.: “According to numerous epidemiological analyses, greater levels of TMAO can disturb the body's metabolic balance and cause a variety of metabolic illnesses, including T2D.” à One should not infer causality from epidemiological studies. Moreover, there is no study cited.
Is there an original work proving the causality of this statement?
- Line 347 ff: “IR changes intestinal environment from glycolytic to proteolytic fermentation…”
Examples for wrong citations:
- Review instead of original publication: lines 263, 267, 290, 329, 333, 460, 465
- Wrong reference: lines 274, 294, 304, 356, 360
- Reference missing: lines 239/240, 253, 351, 380, 386, 426, 440
Examples language weaknesses:
- line 104 “in a bad way”
- line 57 “your risk of getting T2D”
Table 1:
- References completely missing
- Propionate – Microbial agent “Unknown”? There are known propionate producers
- Table legend with abbreviations should be added
- Formatting needed for better readability
See above
Author Response
Responses to the comments and suggestions of Reviewer #1:
- Summary
The manuscript on hand is a narrative review presenting the relevance of microbiome-derived bacterial metabolites in type 2 diabetes. Both animal and human studies and potential intervention approaches aimed at modulating the microbiota and its metabolite production are discussed. The general topic of the review is of general interest, as type 2 diabetes affects a significant number of the populations worldwide, with expected further increasing prevalence. Understanding the pathophysiological mechanisms underlying disease, including those associated with the gut microbiome, is important in the context of therapeutic approaches.
- General concept comments
The aspects recounted here have already been extensively covered in a large number of other reviews and therefore provide little additional understanding beyond previous work. The corresponding original works are often not cited (citation of reviews) or incorrect publications are referenced. In some cases, references are completely missing. In terms of language, the manuscript is at times not appropriate for a scientific publication and warrants major language editing and restructuring of content. It is suggested that the authors address a more specific aspect that has not been addressed as frequently. In doing so, the authors should primarily examine the relevant original work in order to become better acquainted with the topic.
- Specific Comments
Since the overall content should be reconsidered and revised, the following are only exemplary more specific comments on the content.
The authors' conclusions are often too strong and cannot be inferred based on the studies that are provided.
Suggestion 1. Line 314: “Secondary bile acids […] are responsible for the onset of T2D”.
Responses: Thank you for the comments. We have made changes in the original text. (Lines 380-385)
Revised version: These findings suggest that secondary bile acids play a crucial role in regulating body glucose homeostasis and reducing the incidence of T2D through their interactions with receptors such as FXR and TGR5. The activation of these receptors by secondary BA modulates various metabolic processes involved in glucose regulation. Together, these effects contribute to the maintenance of glucose homeostasis and provide potential therapeutic targets for T2D treatment.
Suggestion 2. Line 451: “Therefore, reducing the number of microbial metabolites generated in the gut […] may be a promising approach for treating T2D”.
Responses: Thank you for the comments. We have made changes in the original text. (Lines 500-501)
Revised version: Therefore, modulation of metabolites produced by gut microbes through probiotics as well as drugs can be an effective means of treating T2D[107].
- Li, H. Y.; Zhou, D. D.; Gan, R. Y.; Huang, S. Y.; Zhao, C. N.; Shang, A.; Xu, X. Y.; Li, H. B. Effects and Mechanisms of Probiotics, Prebiotics, Synbiotics, and Postbiotics on Metabolic Diseases Targeting Gut Microbiota: A Narrative Review. Nutrients 2021, 13, doi:10.3390/nu13093211.
Suggestion 3. Line 385 ff.: “According to numerous epidemiological analyses, greater levels of TMAO can disturb the body's metabolic balance and cause a variety of metabolic illnesses, including T2D.” à One should not infer causality from epidemiological studies. Moreover, there is no study cited.
Is there an original work proving the causality of this statement?
Responses: Thank you for your suggestions and pointing out our shortcomings. We have added the relevant elements to the text. (Lines 429-441)
Revised version: According to epidemiological studies, higher levels of TMAO in the body adversely affect T2D[93]. Shan et al. found that in rats with experimentally induced type 2 diabetes, the relative abundance of Clostridiales and Desulfovibrionales, colonies associated with TMAO production, was increased and plasma TMAO levels were elevated in the diabetic group compared with control rats, and promoted atrial inflammation[94]. Elevated serum TMAO levels in T2D patients are strongly associated with cardiovascular risk[95]. In a study of T2D subjects, Adriana et al. found that T2D patients had higher levels of TMAO and an increased risk of NAFLD was observed in T2D patients with elevated levels of TMAO. Elevated levels of TMAO also affected the body's metabolism of circulating BAs[96].
In summary, elevated levels of TMAO have been found to impact the development of T2D to a certain extent. However, the precise mechanism underlying the in-fluence of TMAO on T2D remains unknown, necessitating further exploratory studies in this area.
- Li, S. Y.; Chen, S.; Lu, X. T.; Fang, A. P.; Chen, Y. M.; Huang, R. Z.; Lin, X. L.; Huang, Z. H.; Ma, J. F.; Huang, B. X.; Zhu, H. L. Serum trimethylamine-N-oxide is associated with incident type 2 diabetes in middle-aged and older adults: a prospective cohort study. Journal of translational medicine 2022, 20, 374, doi:10.1186/s12967-022-03581-7.
- Jiang, W. Y.; Huo, J. Y.; Wang, S. C.; Cheng, Y. D.; Lyu, Y. T.; Jiang, Z. X.; Shan, Q. J. Trimethylamine N-oxide facilitates the progression of atrial fibrillation in rats with type 2 diabetes by aggravating cardiac inflammation and connexin remodeling. J Physiol Biochem 2022, 78, 855-867, doi:10.1007/s13105-022-00908-2.
- Su, C.; Li, X.; Yang, Y.; Du, Y.; Zhang, X.; Wang, L.; Hong, B. Metformin alleviates choline diet-induced TMAO elevation in C57BL/6J mice by influencing gut-microbiota composition and functionality. Nutr Diabetes 2021, 11, 27, doi:10.1038/s41387-021-00169-w.
- León-Mimila, P.; Villamil-Ramírez, H.; Li, X. S.; Shih, D. M.; Hui, S. T.; Ocampo-Medina, E.; López-Contreras, B.; Morán-Ramos, S.; Olivares-Arevalo, M.; Grandini-Rosales, P.; Macías-Kauffer, L.; González-González, I.; Hernández-Pando, R.; Gómez-Pérez, F.; Campos-Pérez, F.; Aguilar-Salinas, C.; Larrieta-Carrasco, E.; Villarreal-Molina, T.; Wang, Z.; Lusis, A. J.; Hazen, S. L.; Huertas-Vazquez, A.; Canizales-Quinteros, S. Trimethylamine N-oxide levels are associated with NASH in obese subjects with type 2 diabetes. Diabetes Metab 2021, 47, 101183, doi:10.1016/j.diabet.2020.07.010.
Suggestion 4. Line 347 ff: “IR changes intestinal environment from glycolytic to proteolytic fermentation…”
Responses: Thanks for your correction. We have made corrections in the original text. (lines 391-397)
Revised version: 3-Hydroxyisobutyrate is an intermediate in the catabolism of BCAAs, and in diabetic subjects, increased plasma 3-Hydroxyisobutyrate levels may reflect increased protein catabolism due to relative insulin deficiency resulting from IR-associated insulin secretion deficiency[83]. Inactivation of enzymes involved in BCAAs oxidation was also observed in T2D patients and animal models[84]. In addition, it has been found that improved peripheral insulin sensitivity in T2D patients is associated with a reduction in circulating BCAAs in the body[85].
- Andersson-Hall, U.; Gustavsson, C.; Pedersen, A.; Malmodin, D.; Joelsson, L.; Holmäng, A. Higher Concentrations of BCAAs and 3-HIB Are Associated with Insulin Resistance in the Transition from Gestational Diabetes to Type 2 Diabetes. J Diabetes Res 2018, 2018, 4207067, doi:10.1155/2018/4207067.
- Liu, S.; Li, L.; Lou, P.; Zhao, M.; Wang, Y.; Tang, M.; Gong, M.; Liao, G.; Yuan, Y.; Li, L.; Zhang, J.; Chen, Y.; Cheng, J.; Lu, Y.; Liu, J. Elevated branched-chain α-keto acids exacerbate macrophage oxidative stress and chronic inflammatory damage in type 2 diabetes mellitus. Free Radic Biol Med 2021, 175, 141-154, doi:10.1016/j.freeradbiomed.2021.08.240.
- Chorell, E.; Otten, J.; Stomby, A.; Ryberg, M.; Waling, M.; Hauksson, J.; Svensson, M.; Olsson, T. Improved Peripheral and Hepatic Insulin Sensitivity after Lifestyle Interventions in Type 2 Diabetes Is Associated with Specific Metabolomic and Lipidomic Signatures in Skeletal Muscle and Plasma. Metabolites 2021, 11, doi:10.3390/metabo11120834.
Suggestion 5. Examples for wrong citations:
Review instead of original publication: lines 263, 267, 290, 329, 333, 460, 465
Responses: Thank you for the correction, we have made the correction in the text.
Revised version: (lines 298-302): Chen et al. found that persistent chronic low-grade inflammation in individuals with T2D plays a crucial role in the development of the disease. In T2D, there are elevated levels of inflammatory factors such as IL-6, IL-1β, and TNF-α. The researchers observed that treatment with probiotics resulted in a reduction of these inflammatory factors. Additionally, they discovered that the levels of SCFAs also impact glucose metabolism and IR[66].
- Wang, G.; Si, Q.; Yang, S.; Jiao, T.; Zhu, H.; Tian, P.; Wang, L.; Li, X.; Gong, L.; Zhao, J.; Zhang, H.; Chen, W. Lactic acid bacteria reduce diabetes symptoms in mice by alleviating gut microbiota dysbiosis and inflammation in different manners. Food & function 2020, 11, 5898-5914, doi:10.1039/c9fo02761k.
(lines 304-307): Additionally, studies have shown that agonists of the GPR119 receptor can increase pancreatic β-cell function and insulin production by promoting the release of intestinal GLP-1, leading to a reduction in hyperglycemia[68].
- Matsumoto, K.; Yoshitomi, T.; Ishimoto, Y.; Tanaka, N.; Takahashi, K.; Watanabe, A.; Chiba, K. DS-8500a, an Orally Available G Protein-Coupled Receptor 119 Agonist, Upregulates Glucagon-Like Peptide-1 and Enhances Glucose-Dependent Insulin Secretion and Improves Glucose Homeostasis in Type 2 Diabetic Rats. J Pharmacol Exp Ther 2018, 367, 509-517, doi:10.1124/jpet.118.250019.
(lines 324-329): LPS binds to LPS-binding proteins on macrophages and transmits stimulatory signals through TLR4, located in the lipid bilayer of the cell membrane. This binding initiates an immune response within macrophages. Furthermore, TLR4 activation can induce the production of reactive oxygen species (ROS) and nitric oxide (NO), which contribute to the elimination of invading pathogens and the regulation of immune responses[72].
- Wang, J.; Gao, Y.; Lin, F.; Han, K.; Wang, X. Omentin-1 attenuates lipopolysaccharide (LPS)-induced U937 macrophages activation by inhibiting the TLR4/MyD88/NF-κB signaling. Arch Biochem Biophys 2020, 679, 108187, doi:10.1016/j.abb.2019.108187.
(lines 369-371): Specifically, when secondary bile acids bind to FXR and TGR5, intestinal L cells are stimulated to release the enteric insulinotropic hormone GLP-1, which improves body glucose homeostasis[80].
- Tian, F.; Huang, S.; Xu, W.; Chen, L.; Su, J.; Ni, H.; Feng, X.; Chen, J.; Wang, X.; Huang, Q. Compound K attenuates hyperglycemia by enhancing glucagon-like peptide-1 secretion through activating TGR5 via the remodeling of gut microbiota and bile acid metabolism. J Ginseng Res 2022, 46, 780-789, doi:10.1016/j.jgr.2022.03.006.
(lines 372-374): Mice deficient in FXR have slowed glucose uptake due to inhibition of intrahepatic glycolysis and reduced glucose uptake and utilisation[81].
- Xi, Y.; Li, H. Role of farnesoid X receptor in hepatic steatosis in nonalcoholic fatty liver disease. Biomedicine & pharmacotherapy = Biomedecine & pharmacotherapie 2020, 121, 109609, doi:10.1016/j.biopha.2019.109609.
(lines 511-512): In an animal study, dietary fibre was found to improve lipid metabolism by promoting SCFAs production[109].
- Li, Y.; Xia, D.; Chen, J.; Zhang, X.; Wang, H.; Huang, L.; Shen, J.; Wang, S.; Feng, Y.; He, D.; Wang, J.; Ye, H.; Zhu, Y.; Yang, L.; Wang, W. Dietary fibers with different viscosity regulate lipid metabolism via ampk pathway: roles of gut microbiota and short-chain fatty acid. Poult Sci 2022, 101, 101742, doi:10.1016/j.psj.2022.101742.
(lines 515-517): Studies have shown that dietary fibre obtained from natural vegetables can have an impact on the survival and metabolism of probiotics in the gut and can increase the production of SCFAs, thus benefiting the organism[110].
- Rivas, MÁ; Benito, M. J.; Ruíz-Moyano, S.; Martín, A.; Córdoba, M. G.; Merchán, A. V.; Casquete, R. Improving the Viability and Metabolism of Intestinal Probiotic Bacteria Using Fibre Obtained from Vegetable By-Products. Foods 2021, 10, doi:10.3390/foods10092113.
Wrong reference: lines 274, 294, 304, 356, 360
Revised version: (lines 310-311): Circulating levels of SCFAs were significantly lower in patients with T2D, where propionic acid levels showed a trend of negative correlation with IR[69].
- Salamone, D.; Costabile, G.; Corrado, A.; Della Pepa, G.; Vitale, M.; Giacco, R.; Luongo, D.; Testa, R.; Rivellese, A. A.; Annuzzi, G.; Bozzetto, L. Circulating short-chain fatty acids in type 2 diabetic patients and overweight/obese individuals. Acta diabetologica 2022, 59, 1653-1656, doi:10.1007/s00592-022-01934-8.
(lines 331-334): It was found that patients with T2D had significantly higher levels of LPS, increased levels of Zonulin (ZO-1), a marker of intestinal permeability, and up-regulated levels of the inflammatory factor TNF-α compared to subjects with normal glucose tolerance[73].
- Jayashree, B.; Bibin, Y. S.; Prabhu, D.; Shanthirani, C. S.; Gokulakrishnan, K.; Lakshmi, B. S.; Mohan, V.; Balasubramanyam, M. Increased circulatory levels of lipopolysaccharide (LPS) and zonulin signify novel biomarkers of proinflammation in patients with type 2 diabetes. Mol Cell Biochem 2014, 388, 203-210, doi:10.1007/s11010-013-1911-4.
(lines 340-344): Campa et al. conducted a study on acutely inflamed mice treated with a HFD and LPS, the researchers observed that these mice exhibited several adverse effects, including increased body weight, impaired insulin sensitivity, acute endotoxemia, upregulated expression of the inflammatory marker TLR-4, and impaired glucose homeostasis[76].
- Mendes de Oliveira, E.; Silva, J. C.; Ascar, T. P.; Sandri, S.; Marchi, A. F.; Migliorini, S.; Nakaya, H. T. I.; Fock, R. A.; Campa, A. Acute Inflammation Is a Predisposing Factor for Weight Gain and Insulin Resistance. Pharmaceutics 2022, 14, doi:10.3390/pharmaceutics14030623.
(lines 400-403): Wu et al. found that plasma concentrations of BCAAs affect endothelial function in animals, and the researchers reduced plasma levels of BCAAs and enhanced endothelial cell nitric oxide synthesis and insulin sensitivity in rats by adding AKG (a substrate for BCAA transaminase) to their drinking water[86].
- Tekwe, C. D.; Yao, K.; Lei, J.; Li, X.; Gupta, A.; Luan, Y.; Meininger, C. J.; Bazer, F. W.; Wu, G. Oral administration of α-ketoglutarate enhances nitric oxide synthesis by endothelial cells and whole-body insulin sensitivity in diet-induced obese rats. Experimental biology and medicine (Maywood, N.J.) 2019, 244, 1081-1088, doi:10.1177/1535370219865229.
(lines 404-407): Zhai et al. found that limiting the concentration of BCAAs in HFD maintained blood glucose and insulin at stable levels in mice and prevented HFD-induced obesity, lipid inflammation and IR[87].
- Liu, M.; Huang, Y.; Zhang, H.; Aitken, D.; Nevitt, M. C.; Rockel, J. S.; Pelletier, J. P.; Lewis, C. E.; Torner, J.; Rampersaud, Y. R.; Perruccio, A. V.; Mahomed, N. N.; Furey, A.; Randell, E. W.; Rahman, P.; Sun, G.; Martel-Pelletier, J.; Kapoor, M.; Jones, G.; Felson, D.; Qi, D.; Zhai, G. Restricting Branched-Chain Amino Acids within a High-Fat Diet Prevents Obesity. Metabolites 2022, 12, doi:10.3390/metabo12040334.
Reference missing: lines 239/240, 253, 351, 380, 386, 426, 440
Revised version: (lines 275-279): SCFAs binds to the G protein-coupled receptors free fatty acid receptor 3 (FFAR3) and GPR43 (free fatty acid receptor 2 or FFAR2), which are expressed in a variety of cell types, such as intestinal epithelial cells and enteroendocrine cells[59]. Through GPR41, SCFAs can produce a substance called leptin, which regulates food intake and energy expenditure[60].
- Tolhurst, G.; Heffron, H.; Lam, Y. S.; Parker, H. E.; Habib, A. M.; Diakogiannaki, E.; Cameron, J.; Grosse, J.; Reimann, F.; Gribble, F. M. Short-chain fatty acids stimulate glucagon-like peptide-1 secretion via the G-protein-coupled receptor FFAR2. Diabetes 2012, 61, 364-371, doi:10.2337/db11-1019.
- Xiong, Y.; Miyamoto, N.; Shibata, K.; Valasek, M. A.; Motoike, T.; Kedzierski, R. M.; Yanagisawa, M. Short-chain fatty acids stimulate leptin production in adipocytes through the G protein-coupled receptor GPR41. Proc Natl Acad Sci U S A 2004, 101, 1045-1050, doi:10.1073/pnas.2637002100.
(lines 290-292): Moreover, SCFAs has anti-inflammatory properties and can reduce mucosal and chronic organismal inflammation by inhibiting pro-inflammatory cytokines and promoting the secretion of anti-inflammatory cytokines[64].
- McBride, D. A.; Dorn, N. C.; Yao, M.; Johnson, W. T.; Wang, W.; Bottini, N.; Shah, N. J. Short-chain fatty acid-mediated epigenetic modulation of inflammatory T cells in vitro. Drug Deliv Transl Res 2023, 13, 1912-1924, doi:10.1007/s13346-022-01284-6.
(lines 395-397): In addition, it has been found that improved peripheral insulin sensitivity in T2D patients is associated with a reduction in circulating BCAAs in the body[85].
- Chorell, E.; Otten, J.; Stomby, A.; Ryberg, M.; Waling, M.; Hauksson, J.; Svensson, M.; Olsson, T. Improved Peripheral and Hepatic Insulin Sensitivity after Lifestyle Interventions in Type 2 Diabetes Is Associated with Specific Metabolomic and Lipidomic Signatures in Skeletal Muscle and Plasma. Metabolites 2021, 11, doi:10.3390/metabo11120834.
(lines 422-425): Palika et al. found that patients with T2D with chronic kidney disease had higher serum levels of TMAO, and levels of the inflammatory factors IL-6 and TNF-α were significantly and positively correlated with TMAO levels[92].
- Al-Obaide, M. A. I.; Singh, R.; Datta, P.; Rewers-Felkins, K. A.; Salguero, M. V.; Al-Obaidi, I.; Kottapalli, K. R.; Vasylyeva, T. L. Gut Microbiota-Dependent Trimethylamine-N-oxide and Serum Biomarkers in Patients with T2DM and Advanced CKD. Journal of clinical medicine 2017, 6, doi:10.3390/jcm6090086.
(lines 429-430): According to epidemiological studies, higher levels of TMAO in the body adversely affect T2D[93].
- Li, S. Y.; Chen, S.; Lu, X. T.; Fang, A. P.; Chen, Y. M.; Huang, R. Z.; Lin, X. L.; Huang, Z. H.; Ma, J. F.; Huang, B. X.; Zhu, H. L. Serum trimethylamine-N-oxide is associated with incident type 2 diabetes in middle-aged and older adults: a prospective cohort study. Journal of translational medicine 2022, 20, 374, doi:10.1186/s12967-022-03581-7.
(lines 476-477): Investigations have shown that TMAO increases with increasing levels of IR in humans and animals[102,103].
- Park, J. E.; Miller, M.; Rhyne, J.; Wang, Z.; Hazen, S. L. Differential effect of short-term popular diets on TMAO and other cardio-metabolic risk markers. Nutr Metab Cardiovasc Dis 2019, 29, 513-517, doi:10.1016/j.numecd.2019.02.003.
- Gao, X.; Tian, Y.; Randell, E.; Zhou, H.; Sun, G. Unfavorable Associations Between Serum Trimethylamine N-Oxide and L-Carnitine Levels With Components of Metabolic Syndrome in the Newfoundland Population. Frontiers in endocrinology 2019, 10, 168, doi:10.3389/fendo.2019.00168.
(lines 488-490): By encouraging glycogen metabolism in hepatocytes, the bacterial metabolite sodium butyrate has been found to help maintain glucose levels in balance[104].
- Zhang, W. Q.; Zhao, T. T.; Gui, D. K.; Gao, C. L.; Gu, J. L.; Gan, W. J.; Huang, W.; Xu, Y.; Zhou, H.; Chen, W. N.; Liu, Z. L.; Xu, Y. H. Sodium Butyrate Improves Liver Glycogen Metabolism in Type 2 Diabetes Mellitus. J Agric Food Chem 2019, 67, 7694-7705, doi:10.1021/acs.jafc.9b02083.
Suggestion 6. Examples language weaknesses:
line 104 “in a bad way”
Responses: Thank you for pointing out our shortcomings, we have improved the sentence. (lines 125-128)
Revised version: Chen et al. found that metformin prevented intestinal barrier dysfunction in colitis, attenuated loss of tight junction proteins, and reduced bacterial translocation and levels of the pro-inflammatory factors IL-6 (Interleukin-6), TNF-α (Tumor necrosis factor), and IL-1β (Interleukin-1β)[38].
- Deng, J.; Zeng, L.; Lai, X.; Li, J.; Liu, L.; Lin, Q.; Chen, Y. Metformin protects against intestinal barrier dysfunction via AMPKα1-dependent inhibition of JNK signalling activation. J Cell Mol Med 2018, 22, 546-557, doi:10.1111/jcmm.13342.
line 57 “your risk of getting T2D”
Responses: Thank you for pointing out our shortcomings, we have improved the sentence. (lines 62-64)
Revised version: Bacterial substances such as LPS (lipopolysaccharide), flagellin, and peptidoglycan have been identified as potential causes of an inflammatory reaction, thereby increasing the risk of developing T2D.
Suggestion 7. Table 1:
- References completely missing
Responses: Thank you for the comments. We have added to the table. (line 107)
Revised version:
|
Metabolite |
Microbial agent |
Effect |
Reference |
|
Propionate |
Lactobacillus |
Promotes GLP-1 secretion |
[19] |
|
Butyrate |
Anaerostipes hadrus |
IR |
[20] |
|
isovaleric, lactic acids |
Prevotella copri |
Affects blood sugar levels |
[21] |
|
BA |
Bacteroides fragilis |
Affects blood sugar levels |
[22] |
|
MAM(Microbial anti-inflammatory molecule) |
Faecalibacterium Prausnitzii |
Regulates intestinal barrier function |
[23] |
|
Acetate |
Akkermansia |
Reduce intestinal inflammation |
[24] |
|
SCFA and branched-chain fatty acids |
Acidaminococcus spp, Clostridia spp. |
Amino acid fermentation |
[25][26] |
|
Tryptamine |
Ruminococcus gnavus |
IR |
[27][28] |
|
Histamine |
Morganella morganii |
Decarboxylation of histidine (histidine decarboxylase (HDC)) |
[29][30] |
|
Imidazole propionate (ImP) |
Adlercreutziae equolifaciens Anaerococcus Prevotii |
Glucose tolerance |
[31] |
|
Dopamine |
Enterococcus faecalis |
Insulin sensitivity |
[32][33] |
- Wang, Y.; Dilidaxi, D.; Wu, Y.; Sailike, J.; Sun, X.; Nabi, X. H. Composite probiotics alleviate type 2 diabetes by regulating intestinal microbiota and inducing GLP-1 secretion in db/db mice. Biomedicine & pharmacotherapy = Biomedecine & pharmacotherapie 2020, 125, 109914, doi:10.1016/j.biopha.2020.109914.
- Frias, J. P.; Lee, M. L.; Carter, M. M.; Ebel, E. R.; Lai, R. H.; Rikse, L.; Washington, M. E.; Sonnenburg, J. L.; Damman, C. J. A microbiome-targeting fibre-enriched nutritional formula is well tolerated and improves quality of life and haemoglobin A1c in type 2 diabetes: A double-blind, randomized, placebo-controlled trial. Diabetes Obes Metab 2023, 25, 1203-1212, doi:10.1111/dom.14967.
- Liu, J.; Zhou, L.; Sun, L.; Ye, X.; Ma, M.; Dou, M.; Shi, L. Association Between Intestinal Prevotella copri Abundance and Glycemic Fluctuation in Patients with Brittle Diabetes. Diabetes, metabolic syndrome and obesity : targets and therapy 2023, 16, 1613-1621, doi:10.2147/dmso.S412872.
- Mastropaolo, M. D.; Evans, N. P.; Byrnes, M. K.; Stevens, A. M.; Robertson, J. L.; Melville, S. B. Synergy in polymicrobial infections in a mouse model of type 2 diabetes. Infect Immun 2005, 73, 6055-6063, doi:10.1128/iai.73.9.6055-6063.2005.
- Xu, J.; Liang, R.; Zhang, W.; Tian, K.; Li, J.; Chen, X.; Yu, T.; Chen, Q. Faecalibacterium prausnitzii-derived microbial anti-inflammatory molecule regulates intestinal integrity in diabetes mellitus mice via modulating tight junction protein expression. Journal of diabetes 2020, 12, 224-236, doi:10.1111/1753-0407.12986.
- Zhang, W.; Xu, J. H.; Yu, T.; Chen, Q. K. Effects of berberine and metformin on intestinal inflammation and gut microbiome composition in db/db mice. Biomedicine & pharmacotherapy = Biomedecine & pharmacotherapie 2019, 118, 109131, doi:10.1016/j.biopha.2019.109131.
- Deehan, E. C.; Yang, C.; Perez-Muñoz, M. E.; Nguyen, N. K.; Cheng, C. C.; Triador, L.; Zhang, Z.; Bakal, J. A.; Walter, J. Precision Microbiome Modulation with Discrete Dietary Fiber Structures Directs Short-Chain Fatty Acid Production. Cell host & microbe 2020, 27, 389-404.e386, doi:10.1016/j.chom.2020.01.006.
- Smith, E. A.; Macfarlane, G. T. Dissimilatory amino Acid metabolism in human colonic bacteria. Anaerobe 1997, 3, 327-337, doi:10.1006/anae.1997.0121.
- Wang, Y.; Ye, X.; Ding, D.; Lu, Y. Characteristics of the intestinal flora in patients with peripheral neuropathy associated with type 2 diabetes. J Int Med Res 2020, 48, 300060520936806, doi:10.1177/0300060520936806.
- Williams, B. B.; Van Benschoten, A. H.; Cimermancic, P.; Donia, M. S.; Zimmermann, M.; Taketani, M.; Ishihara, A.; Kashyap, P. C.; Fraser, J. S.; Fischbach, M. A. Discovery and characterization of gut microbiota decarboxylases that can produce the neurotransmitter tryptamine. Cell host & microbe 2014, 16, 495-503, doi:10.1016/j.chom.2014.09.001.
- Valles-Colomer, M.; Falony, G.; Darzi, Y.; Tigchelaar, E. F.; Wang, J.; Tito, R. Y.; Schiweck, C.; Kurilshikov, A.; Joossens, M.; Wijmenga, C.; Claes, S.; Van Oudenhove, L.; Zhernakova, A.; Vieira-Silva, S.; Raes, J. The neuroactive potential of the human gut microbiota in quality of life and depression. Nat Microbiol 2019, 4, 623-632, doi:10.1038/s41564-018-0337-x.
- Barcik, W.; Wawrzyniak, M.; Akdis, C. A.; O'Mahony, L. Immune regulation by histamine and histamine-secreting bacteria. Curr Opin Immunol 2017, 48, 108-113, doi:10.1016/j.coi.2017.08.011.
- Koh, A.; Molinaro, A.; Ståhlman, M.; Khan, M. T.; Schmidt, C.; Mannerås-Holm, L.; Wu, H.; Carreras, A.; Jeong, H.; Olofsson, L. E.; Bergh, P. O.; Gerdes, V.; Hartstra, A.; de Brauw, M.; Perkins, R.; Nieuwdorp, M.; Bergström, G.; Bäckhed, F. Microbially Produced Imidazole Propionate Impairs Insulin Signaling through mTORC1. Cell 2018, 175, 947-961.e917, doi:10.1016/j.cell.2018.09.055.
- Tavares, G.; Marques, D.; Barra, C.; Rosendo-Silva, D.; Costa, A.; Rodrigues, T.; Gasparini, P.; Melo, B. F.; Sacramento, J. F.; Seiça, R.; Conde, S. V.; Matafome, P. Dopamine D2 receptor agonist, bromocriptine, remodels adipose tissue dopaminergic signalling and upregulates catabolic pathways, improving metabolic profile in type 2 diabetes. Mol Metab 2021, 51, 101241, doi:10.1016/j.molmet.2021.101241.
- Maini Rekdal, V.; Bess, E. N.; Bisanz, J. E.; Turnbaugh, P. J.; Balskus, E. P. Discovery and inhibition of an interspecies gut bacterial pathway for Levodopa metabolism. Science (New York, N.Y.) 2019, 364, doi:10.1126/science.aau6323.
- Propionate – Microbial agent “Unknown”? There are known propionate producers
Responses: Thank you for the correction, we have added it to the table.
Revised version: Microbial agent is Lactobacillus.
- Table legend with abbreviations should be added
Responses: Thanks to your suggestion, we have added a list of abbreviations to the text. (line 46)
Revised version:
|
Abbreviations |
|
Abbreviations |
|
Abbreviations |
|
|
T2D |
Type 2 diabetes |
AMP |
Anti-microbial peptides |
CDCA |
Chenodeoxycholic acid |
|
SCFA |
short-chain fatty acids |
IgA |
Immunoglobulin A |
CYP7A1 |
Cholesterol-7α-hydroxylase |
|
GPR |
G prtein-coupled receptor |
TJ |
Tight junctions |
CYP27A1 |
Sterol 27-hydroxylase |
|
GLP |
glucagon-like peptide |
IgG |
Immunoglobulin G |
TCDCA |
Tauro chenodeoxycholic acid |
|
PYY |
peptide YY |
GLUT2 |
Glucose transporter protein |
TCA |
Taurocholic acid |
|
TMA |
Trimethylamine |
HFD |
High-fat diet |
GCA |
Glycocholic acid |
|
TMAO |
Trimethylamine N-Oxide |
PAMPs |
Pathogen-associated molecular patterns |
GCDCA |
Glycochenodeoxycholicacid |
|
LPS |
lipopolysaccharide |
IR |
Insulin resistance |
BSH |
Bile salt hydrolases |
|
MAM |
Microbial anti-inflammatory molecule |
PPARγ |
Peroxisome proliferator-activated receptor γ |
DCA |
Deoxycholic acid |
|
BA |
Bile acids |
iNOS |
inducible nitric oxide synthase |
LCA |
Lithocholic acid |
|
IL-6 |
Interleukin-6 |
FFAR3 |
G protein-coupled receptors free fatty acid receptor 3 |
UDCA |
Ursodeoxycholic acid |
|
TNF-α |
Tumor necrosis factor |
GPR43 |
Free fatty acid receptor 2 |
FXR |
Farnesoid X receptor |
|
IL-1β |
Interleukin-1β |
TLRs |
Toll-like receptors |
VDR |
Vitamin D receptor |
|
BCAAs |
Branched-chain amino acids |
NLRP3 |
NOD-like receptor family pyrin domain containing 3 |
PXR |
Pregnane X receptor |
|
GPCR |
G protein-coupled receptor |
ZO-1 |
Zonulin |
TGR5 |
G protein-coupled receptor 5 |
|
IL-1 |
Interleukin-1 |
CA |
Cholic acid |
FMO3 |
Flavin-containing monooxygenase 3 |
- Formatting needed for better readability
Responses: Thank you for your suggestion, we have improved the table format. (line 107)

Reviewer 2 Report
· The abstract is very short. Please define the process for conducting a literature search (e.g., we used pubmed/Medline to.....). Why this review is important? What is a gap in a review topic? The conclusion should propose a clear direction for future studies.
· Line 30: Please clarify. “a sedentary lifestyle” e.g., diet, physical activity.
· Could you please clarify the differences between SCFAs and branched chain-FA in the paper?
· The novelty aspect of the review is particularly weak. This should be clearly articulated at the end of introduction. My main concern is several reviews on such topic are performed (Front Endocrinol (Lausanne). 2023 May 9;14:1114424; Biomed Pharmacother. 2022 May;149:112839; Mediators Inflamm. 2021 Aug 17;2021:5110276; Front Cell Infect Microbiol. 2022 Feb 15;12:834485; Curr Nutr Rep. 2020 Jun;9(2):83-93). I would suggest that the authors to present the aim of the paper with regards to what is currently known by previous reviews, therefore highlighting the added value of this study. In my opinion, this review adds nothing in light of previous reviews.
· In introduction, please include if there is any evidence investigating the role of gut metabolites as potential mechanisms that may have therapeutic implications for reducing different types of diabetes. I would suggest authors referring to these articles (Pediatr Diabetes. 2021 May;22(3):425-433; EClinicalMedicine. 2023 Aug 3;62:102132; Genes (Basel). 2023 Apr 29;14(5):1017. Nutrients. 2022 Sep; 14(18): 3727).
· Although this is a narrative review, the process to literature searching should be clearly mentioned. Please include one paragraph clarifying the criteria adopted to perform the search strategy. The method section should include the type of articles, search terms, publication date (the literature search cover), databases and inclusion/exclusion criteria.
· Table 1 would benefit from including more columns. It is vague in its current format.
· Line 122, 196: Is there any evidence of other types of diet that may induce gut dysbiosis/inflammation in diabetes?
· Several sentences are not supported with any references (e.g., Line 247-280). Please also check throughout the paper.
· I would suggest authors to add two tables to provide a visual overview of human and animals studies in section 5.
· The conclusion is very short in its current format and should be expanded to provide more details about the results of the review. Future implications for future studies should be clearly mentioned.
Moderate revisions required.
Author Response
Responses to the comments and suggestions of Reviewer #2:
Suggestion 1. The abstract is very short. Please define the process for conducting a literature search (e.g., we used pubmed/Medline to.....). Why this review is important? What is a gap in a review topic? The conclusion should propose a clear direction for future studies.
Responses: Thank you for your suggestion, we have added the process of literature search in the text. (lines 583-588)
Revised version: As of September 2023, we conducted a literature search in Pubmed and web of science data-bases, mostly searching for articles published in English in the last 5 years, including reviews and research-based papers. The following keywords were searched: "gut microbial metabolites and T2D", "short-chain fatty acids", "bile acids", "branched-chain amino acids", "intestinal epithelial barrier dysfunction". To study the effect of gut microbial metabolites on T2D.
Why this review is important?
Responses: Thank you for your suggestions, which we have added to the text. (lines 19-20)
Revised version: A deeper understanding of the link between gut microbial metabolites and T2D will enhance our knowledge of the disease and may offer new treatment approaches.
What is a gap in a review topic?
Responses: Thank you for your suggestions, which we have added to the text. (lines 20-23)
Revised version: Although many animal studies have investigated the palliative and attenuating effects of gut microbial metabolites on T2D, few have established a complete cure. Therefore, conducting more systematic studies in the future is necessary.
The conclusion should propose a clear direction for future studies.
Responses: Thank you for your suggestions, which we have added to the conclusions section. (lines 574-579)
Revised version: So, we think this can provide a thought for future researchers. Since gut microbial metabolites have an important role in T2D, there is a need to develop a targeted therapeutic agent based on gut microbial metabolites in the future. In addition, because of the large number of gut microbial species, the exact mechanism of which specific gut microbes treat T2D is unknown, and the question of whether there are other undiscovered metabolites that may have an impact on T2D remains to be examined.
Suggestion 2. Line 30: Please clarify. “a sedentary lifestyle” e.g., diet, physical activity.
Responses: Thank you for your suggestion, we have corrected it in the text. (lines 34-36)
Revised version: Although genetic susceptibility is a major contributor to T2D, physical inactivity lifestyle[3] and obesity are also important contributors to T2D[4].
Suggestion 3. Could you please clarify the differences between SCFAs and branched chain-FA in the paper?
Responses: Thank you for your suggestion.
We explain the difference as follows: Bacteria in the colon and cecum produce SCFA, a class of carboxylic acids with only 1-6 carbon atoms, by fermenting undigested dietary fibre; the main SCFA are acetic, propionic and butyric acids[1]. To some extent, bacteria produce branched-chain fatty acids, such as isobutyric and isovaleric acids[2], through protein fermentation. Branched-chain fatty acids are mainly saturated fatty acids, which occur naturally in bacterial lipids and several fungi, and can also be synthesised by rumen microbes in ruminants[3].
Reference:
- Du, L.; Li, Q.; Yi, H.; Kuang, T.; Tang, Y.; Fan, G. Gut microbiota-derived metabolites as key actors in type 2 diabetes mellitus. Biomedicine & pharmacotherapy = Biomedecine & pharmacotherapie 2022, 149, 112839, doi:10.1016/j.biopha.2022.112839.
- Trefflich, I.; Dietrich, S.; Braune, A.; Abraham, K.; Weikert, C. Short- and Branched-Chain Fatty Acids as Fecal Markers for Microbiota Activity in Vegans and Omnivores. Nutrients 2021, 13, doi:10.3390/nu13061808.
- Maheshwari, G.; Ringseis, R.; Wen, G.; Gessner, D. K.; Rost, J.; Fraatz, M. A.; Zorn, H.; Eder, K. Branched-Chain Fatty Acids as Mediators of the Activation of Hepatic Peroxisome Proliferator-Activated Receptor Alpha by a Fungal Lipid Extract. Biomolecules 2020, 10, doi:10.3390/biom10091259.
Suggestion 4. The novelty aspect of the review is particularly weak. This should be clearly articulated at the end of introduction. My main concern is several reviews on such topic are performed (Front Endocrinol (Lausanne). 2023 May 9;14:1114424; Biomed Pharmacother. 2022 May;149:112839; Mediators Inflamm. 2021 Aug 17;2021:5110276; Front Cell Infect Microbiol. 2022 Feb 15;12:834485; Curr Nutr Rep. 2020 Jun;9(2):83-93). I would suggest that the authors to present the aim of the paper with regards to what is currently known by previous reviews, therefore highlighting the added value of this study. In my opinion, this review adds nothing in light of previous reviews.
Responses: We thank you for your advice on the revision of our article. We have carefully reviewed several of the reviews you recommended, and based on what is known from previous reviews, we have added our reflections on the present review, suggesting shortcomings in the currently available research papers.
Revised version: (lines 19-23): A deeper understanding of the link between gut microbial metabolites and T2D will enhance our knowledge of the disease and may offer new treatment approaches. Although many animal studies have investigated the palliative and attenuating effects of gut microbial metabolites on T2D, few have established a complete cure. Therefore, conducting more systematic studies in the future is necessary.
What we think is novel about the article. (lines 98-101): We found that most of the review literature provided few summary overviews of the relevant studies that have been carried out, so we have provided a summary overview of the relevant studies in Section 5 of the article.
Suggestion 5. In introduction, please include if there is any evidence investigating the role of gut metabolites as potential mechanisms that may have therapeutic implications for reducing different types of diabetes. I would suggest authors referring to these articles (Pediatr Diabetes. 2021 May;22(3):425-433; EClinicalMedicine. 2023 Aug 3;62:102132; Genes (Basel). 2023 Apr 29;14(5):1017. Nutrients. 2022 Sep; 14(18): 3727).
Responses: Thank you for your suggestions, which we have added to the text. (lines 81-95)
Revised version: Gestational diabetes is associated with dysbiosis of the gut microbiota in newborns, according to Naser A. Alsharairi's study, which also found that the gut microbial metabolite SCFAs affects the expression of diabetes-related genes in newborns[15]. Huixia Yang et al. discovered that acetic, propionic, and butyric acids have potential antidiabetic and anti-inflammatory effects in women with gestational diabetes, as demonstrated by their levels in maternal circulation[16]. These SCFAs could be a therapeutic target for treating gestational diabetes. Furthermore, according to a study by Jessica E. Harbison et al., lower levels of circulating SCFAs are also associated with type 1 diabetes (T1D) in youth[17], which emphasizes the potential therapeutic role of gut microbial metabolites, including SCFAs, in different types of diabetes. Xiao et al. found low levels of phenolic acids and SCFAs in patients with T1D. Phenolic acids have demonstrated antidiabetic and anti-inflammatory effects. The reduced production of SCFAs in T1D patients could be attributed to the decreased abundance of SCFAs-producing flora resulting from the loss of phenolic acids[18]. Therefore, supplementing with phenolic acid compounds may be an effective approach for treating T1D.
- Alsharairi, N. A. Exploring the Diet-Gut Microbiota-Epigenetics Crosstalk Relevant to Neonatal Diabetes. Genes 2023, 14, doi:10.3390/genes14051017.
- Wang, S.; Liu, Y.; Qin, S.; Yang, H. Composition of Maternal Circulating Short-Chain Fatty Acids in Gestational Diabetes Mellitus and Their Associations with Placental Metabolism. Nutrients 2022, 14, doi:10.3390/nu14183727.
- Harbison, J. E.; Thomson, R. L.; Wentworth, J. M.; Louise, J.; Roth-Schulze, A.; Battersby, R. J.; Ngui, K. M.; Penno, M. A. S.; Colman, P. G.; Craig, M. E.; Barry, S. C.; Tran, C. D.; Makrides, M.; Harrison, L. C.; Couper, J. J. Associations between diet, the gut microbiome and short chain fatty acids in youth with islet autoimmunity and type 1 diabetes. Pediatr Diabetes 2021, 22, 425-433, doi:10.1111/pedi.13178.
- Hu, J.; Ding, J.; Li, X.; Li, J.; Zheng, T.; Xie, L.; Li, C.; Tang, Y.; Guo, K.; Huang, J.; Liu, S.; Yan, J.; Peng, W.; Hou, C.; Wen, L.; Xu, A.; Zhou, Z.; Xiao, Y. Distinct signatures of gut microbiota and metabolites in different types of diabetes: a population-based cross-sectional study. EClinicalMedicine 2023, 62, 102132, doi:10.1016/j.eclinm.2023.102132.
Suggestion 6. Although this is a narrative review, the process to literature searching should be clearly mentioned. Please include one paragraph clarifying the criteria adopted to perform the search strategy. The method section should include the type of articles, search terms, publication date (the literature search cover), databases and inclusion/exclusion criteria.
Responses: Thank you for your suggestions, which we have added to the text accordingly. (lines 583-588)
Revised version: As of September 2023, we conducted a literature search in Pubmed and web of science data-bases, mostly searching for articles published in English in the last 5 years, including reviews and research-based papers. The following keywords were searched: "gut microbial metabolites and T2D", "short-chain fatty acids", "bile acids", "branched-chain amino acids ", "intestinal epithelial barrier dysfunction". To study the effect of gut microbial metabolites on T2D.
Suggestion 7. Table 1 would benefit from including more columns. It is vague in its current format.
Responses: Thank you for your suggestion, we have adjusted the form. (line 107)
Revised version:
|
Metabolite |
Microbial agent |
Effect |
Reference |
|
Propionate |
Lactobacillus |
Promotes GLP-1 secretion |
[19] |
|
Butyrate |
Anaerostipes hadrus |
IR |
[20] |
|
isovaleric, lactic acids |
Prevotella copri |
Affects blood sugar levels |
[21] |
|
BA |
Bacteroides fragilis |
Affects blood sugar levels |
[22] |
|
MAM(Microbial anti-inflammatory molecule) |
Faecalibacterium Prausnitzii |
Regulates intestinal barrier function |
[23] |
|
Acetate |
Akkermansia |
Reduce intestinal inflammation |
[24] |
|
SCFA and branched-chain fatty acids |
Acidaminococcus spp, Clostridia spp. |
Amino acid fermentation |
[25][26] |
|
Tryptamine |
Ruminococcus gnavus |
IR |
[27][28] |
|
Histamine |
Morganella morganii |
Decarboxylation of histidine (histidine decarboxylase (HDC)) |
[29][30] |
|
Imidazole propionate (ImP) |
Adlercreutziae equolifaciens Anaerococcus Prevotii |
Glucose tolerance |
[31] |
|
Dopamine |
Enterococcus faecalis |
Insulin sensitivity |
[32][33] |
Suggestion 8. Line 122, 196: Is there any evidence of other types of diet that may induce gut dysbiosis/inflammation in diabetes?
Responses: Thank you for your suggestions, which we have added to the text. (lines 146-156, lines 227-234)
Revised version: (lines 146-156): Farid Najafi et al. conducted a study that highlights the association between Western diets, which include red meat and processed meats, and the pathogenesis of inflammatory diseases, their findings demonstrated that individuals with T2D have significantly higher pro-inflammatory dietary intake than non-T2D, elevated levels of the inflammatory factors IL-1 (Interleukin-1) and TNF-α result from this type of diet, these factors can interfere with insulin signaling, leading to the development of insulin resistance[45]. Demirer et al. conducted a study that investigated the effects of dietary advanced glycation end products (AGEs) on inflammation, their findings suggest that the intake of dietary AGEs can directly or indirectly induce inflammation, given that individuals with diabetes are prone to oxidative stress and inflammation, excessive consumption of dietary AGEs may expedite the inflammatory process[46].
(lines 227-234): Hu et al. showed that a high-sugar diet induced symptoms of T2D in rats, causing inflammation, disturbances in glucose metabolism and lipid metabolism, as well as a higher abundance of harmful bacteria present in the gut[55]. Guo et al. found that a Methionine/choline deficient diet resulted in the expression of pro-inflammatory factors in mice, while a Methionine/choline deficient diet resulted in elevated serum insulin levels in mice with symptoms similar to those of T2D[56]. Estaphan et al. found that a diet high in iron caused insulin damage and inflammatory cell infiltration in rats, increasing the risk of developing T2D[57].
- Namazi, N.; Anjom-Shoae, J.; Najafi, F.; Ayati, M. H.; Darbandi, M.; Pasdar, Y. Pro-inflammatory diet, cardio-metabolic risk factors and risk of type 2 diabetes: A cross-sectional analysis using data from RaNCD cohort study. BMC Cardiovasc Disord 2023, 23, 5, doi:10.1186/s12872-022-03023-8.
- Demirer, B.; Yardımcı, H.; Erem Basmaz, S. Inflammation level in type 2 diabetes is associated with dietary advanced glycation end products, Mediterranean diet adherence and oxidative balance score: A pathway analysis. J Diabetes Complications 2023, 37, 108354, doi:10.1016/j.jdiacomp.2022.108354.
- Zhao, R.; Li, N.; Liu, W.; Liu, Q.; Zhang, L.; Peng, X.; Zhao, R.; Hu, H. Low glycemic index potato biscuits alleviate physio-histological damage and gut dysbiosis in rats with type-2 diabetes mellitus induced by high-sugar and high-fat diet and streptozotocin. The Journal of nutritional biochemistry 2023, 119, 109401, doi:10.1016/j.jnutbio.2023.109401.
- Zhen, Q.; Liang, Q.; Wang, H.; Zheng, Y.; Lu, Z.; Bian, C.; Zhao, X.; Guo, X. Theabrownin ameliorates liver inflammation, oxidative stress, and fibrosis in MCD diet-fed C57BL/6J mice. Frontiers in endocrinology 2023, 14, 1118925, doi:10.3389/fendo.2023.1118925.
- Delghingaro-Augusto, V.; Hosaka, A.; Estaphan, S.; Richardson, A.; Dahlstrom, J. E.; Nolan, C. J. High Dietary Iron in Western Diet-Fed Male Rats Causes Pancreatic Islet Injury and Acute Pancreatitis. J Nutr 2023, 153, 723-732, doi:10.1016/j.tjnut.2023.01.009.
Suggestion 9. Several sentences are not supported with any references (e.g., Line 247-280). Please also check throughout the paper.
Responses: Thank you for your suggestions and I apologize for our carelessness. We have added to the text. (lines 291-293, lines 299-304, lines 306-308, lines 312-313)
Revised version: (lines 290-292): Moreover, SCFAs has anti-inflammatory properties and can reduce mucosal and chronic organismal inflammation by inhibiting pro-inflammatory cytokines and promoting the secretion of anti-inflammatory cytokines[64].
(lines 298-302): Chen et al. found that persistent chronic low-grade inflammation in individuals with T2D plays a crucial role in the development of the disease. In T2D, there are elevated levels of inflammatory factors such as IL-6, IL-1β, and TNF-α. The researchers observed that treatment with probiotics resulted in a reduction of these inflammatory factors. Additionally, they discovered that the levels of SCFAs also impact glucose metabolism and IR[66].
(lines 304-307): Additionally, studies have shown that agonists of the GPR119 receptor can increase pancreatic beta-cell function and insulin production by promoting the release of intestinal GLP-1, leading to a reduction in hyperglycemia[68].
(lines 310-311): Circulating levels of SCFAs were significantly lower in patients with T2D, where propionic acid levels showed a trend of negative correlation with IR[69].
- McBride, D. A.; Dorn, N. C.; Yao, M.; Johnson, W. T.; Wang, W.; Bottini, N.; Shah, N. J. Short-chain fatty acid-mediated epigenetic modulation of inflammatory T cells in vitro. Drug Deliv Transl Res 2023, 13, 1912-1924, doi:10.1007/s13346-022-01284-6.
- Wang, G.; Si, Q.; Yang, S.; Jiao, T.; Zhu, H.; Tian, P.; Wang, L.; Li, X.; Gong, L.; Zhao, J.; Zhang, H.; Chen, W. Lactic acid bacteria reduce diabetes symptoms in mice by alleviating gut microbiota dysbiosis and inflammation in different manners. Food & function 2020, 11, 5898-5914, doi:10.1039/c9fo02761k.
- Matsumoto, K.; Yoshitomi, T.; Ishimoto, Y.; Tanaka, N.; Takahashi, K.; Watanabe, A.; Chiba, K. DS-8500a, an Orally Available G Protein-Coupled Receptor 119 Agonist, Upregulates Glucagon-Like Peptide-1 and Enhances Glucose-Dependent Insulin Secretion and Improves Glucose Homeostasis in Type 2 Diabetic Rats. J Pharmacol Exp Ther 2018, 367, 509-517, doi:10.1124/jpet.118.250019.
- Salamone, D.; Costabile, G.; Corrado, A.; Della Pepa, G.; Vitale, M.; Giacco, R.; Luongo, D.; Testa, R.; Rivellese, A. A.; Annuzzi, G.; Bozzetto, L. Circulating short-chain fatty acids in type 2 diabetic patients and overweight/obese individuals. Acta diabetologica 2022, 59, 1653-1656, doi:10.1007/s00592-022-01934-8.
Suggestion 10. I would suggest authors to add two tables to provide a visual overview of human and animals studies in section 5.
Responses: Thank you for your suggestions, which we have added to the text accordingly. (line 505, line 553)
Revised version:
(line 526) Table 2
|
Exogenous substances affecting metabolites |
Regulated substance |
Symptoms of improvement |
Reference |
|
Poria oligosaccharides |
BA |
glucose intolerance, and insulin resistance |
[97] |
|
Berberine and Metformin |
LPS |
Regulates blood sugar levels and reduces intestinal inflammation |
[24] |
|
Legumes Diet |
Gut microorganisms |
Reduce serum total cholesterol and fasting blood glucose levels |
[98][99] |
|
Sodium butyrate |
glycogen |
Maintenance of glucose homeostasis |
[104] |
|
Extract of Sargarsum fusiforme |
BCAAs |
Alleviation of metabolic disorders |
[105] |
|
Probiotics |
Intestinal proinsulin |
Insulin sensitivity |
[19] |
|
Probiotics and Prebiotics |
Gut microbial metabolites |
Alleviation of T2D |
[100] |
(line 575) Table 3
|
Exogenous substances affecting metabolites |
Regulated metabolites |
Impact on T2D |
Reference |
|
Dietary fibre |
SCFAs |
Improvement of glucose metabolism |
[110] |
|
Fruits and grains |
SCFAs |
Reduces inflammation and maintains intestinal mucosal homeostasis |
[111] |
|
Excessive intake of protein |
BCAAs |
Interferes with metabolism and elevates risk of insulin resistance |
[112][113] |
|
Red Meat Foods |
TMAO |
Interfering with insulin-related signalling pathways |
[114] |
|
Diet |
Indole-propionic acid |
Stabilisation of insulin secretion |
[116] |
Suggestion 11. The conclusion is very short in its current format and should be expanded to provide more details about the results of the review. Future implications for future studies should be clearly mentioned.
Responses: Thank you for your suggestions, which we have added to the conclusions section. (lines 563-573, lines 574-579)
Revised version: (lines 563-573): Patients with T2D have abnormal insulin secretion, disturbed glucose metabolism as well as develop IR. Gut microbial metabolites SCFAs can modulate low-level inflammatory responses and improve glucose metabolism. BAs have the ability to interact with receptors such as FXR and VDR, allowing them to regulate glucose homeostasis in both the gut and liver. On the other hand, high levels of LPS can lead to increased intestinal permeability, resulting in bacteraemia and impairing insulin signaling pathways. Furthermore, abnormal levels of BCAAs have been associated with an increased risk of metabolic disorders in the organism. Additionally, elevated levels of TMAO have been linked to the development of IR. To effectively treat T2D, it is crucial to maximize the beneficial role of metabolites like BAs and SCFAs, while simultaneously reducing the levels of potentially harmful metabolites, namely LPS, BCAA, and TMAO.
(lines 574-579): So, we think this can provide a thought for future researchers. Since gut microbial metabolites have an important role in T2D, there is a need to develop a targeted therapeutic agent based on gut microbial metabolites in the future. In addition, because of the large number of gut microbial species, the exact mechanism of which specific gut microbes treat T2D is unknown, and the question of whether there are other undiscovered metabolites that may have an impact on T2D remains to be examined.

Reviewer 3 Report
In this review, the authors have summarized the link between gut microbiota, microbial metabolites, and T2D. There are several reviews on this subject, however, the authors have done a good job of adding schematics/figures for easier understanding. Lines 482-483 state consumption of indole-propionic acid is linked to a higher incidence of T2D, but the cited paper (76) indicates a negative correlation between IPA and developing T2DM. Lines 486-489 also state a protective effect of IPA on T2D development.
Author Response
Responses to the comments and suggestions of Reviewer #3:
Suggestion 1. In this review, the authors have summarized the link between gut microbiota, microbial metabolites, and T2D. There are several reviews on this subject, however, the authors have done a good job of adding schematics/figures for easier understanding. Lines 482-483 state consumption of indole-propionic acid is linked to a higher incidence of T2D, but the cited paper (76) indicates a negative correlation between IPA and developing T2DM. Lines 486-489 also state a protective effect of IPA on T2D development.
Responses: Thank you for the comments. After reviewing the literature, we realised that this sentence was incorrect and we have deleted it from the text.
Revised version: (lines 533-534): The association between Bacillus butyricus and remission of insulin resistance was particularly striking[115].
(lines 534-537): Additionally, in a study of diabetic individuals from Finland, indole-propionic acid was found to have a protective have an effect for the development of T2D. This protective effect may be due to indolepropionic acid's ability to control enteroendocrine L-cell production, which preserves insulin secretion[116].
- Wang, M.; Li, L.; Chen, Y.; Lian, G.; Wang, J.; Zhang, J.; Shan, K.; Shang, L.; Tian, F.; Jing, C. Role of Gut Microbiome and Microbial Metabolites in Alleviating Insulin Resistance After Bariatric Surgery. Obesity surgery 2021, 31, 327-336, doi:10.1007/s11695-020-04974-7.
116. de Mello, V. D.; Paananen, J.; Lindström, J.; Lankinen, M. A.; Shi, L.; Kuusisto, J.; Pihlajamäki, J.; Auriola, S.; Lehtonen, M.; Rolandsson, O.; Bergdahl, I. A.; Nordin, E.; Ilanne-Parikka, P.; Keinänen-Kiukaanniemi, S.; Landberg, R.; Eriksson, J. G.; Tuomilehto, J.; Hanhineva, K.; Uusitupa, M. Indolepropionic acid and novel lipid metabolites are associated with a lower risk of type 2 diabetes in the Finnish Diabetes Prevention Study. Scientific reports 2017, 7, 46337, doi:10.1038/srep46337.

Reviewer 4 Report
In the manuscript submitted to me for review entitled " A metabolite perspective on the involvement of the gut microbiota in type 2 diabetes“ the authors Yifeng Fu, Siying Li, Yunhua Xiao, Jun Fang, Gang Liu summarize data for the role of microbiota metabolites in influencing the development of type 2 diabetes and provide various pharmacological and dietary options that could serve as therapeutics to reduce the risk of developing diabetes.
The study was carried out extremely thoroughly based on studies presented in 80 references, examining the problem from the last 15 years. 60 of the cited references are from the last 5 years (3/4 of the total number).
The presented results are supported visually with 5 figures and 1 table, which are extremely well laid out and illustrated.
I have absolutely no comments on the main part of the manuscript, the information is presented in a very detailed and orderly manner and I believe it will be of great interest to the reader. But since the manuscript is of the review type, I have a few remarks about the References section, since such articles usually receive a lot of attention and are highly cited, everything in this section should be perfect.
1. In some of the references not all authors are indicated (No. 9, 23, 26, 33, 34, 35, 51, 60, 68 and 77), let them all be added.
2. In reference #18, the names of the authors are abbreviated somewhat too much. Let it be checked and corrected.
3. My personal opinion is that given that we are in September of 2023, it would be very valuable for the manuscript if there were at least a few references with information from the current year. I'm sure there should be one.
Author Response
Responses to the comments and suggestions of Reviewer #4:
In the manuscript submitted to me for review entitled " A metabolite perspective on the involvement of the gut microbiota in type 2 diabetes“ the authors Yifeng Fu, Siying Li, Yunhua Xiao, Jun Fang, Gang Liu summarize data for the role of microbiota metabolites in influencing the development of type 2 diabetes and provide various pharmacological and dietary options that could serve as therapeutics to reduce the risk of developing diabetes.
The study was carried out extremely thoroughly based on studies presented in 80 references, examining the problem from the last 15 years. 60 of the cited references are from the last 5 years (3/4 of the total number).
The presented results are supported visually with 5 figures and 1 table, which are extremely well laid out and illustrated.
I have absolutely no comments on the main part of the manuscript, the information is presented in a very detailed and orderly manner and I believe it will be of great interest to the reader. But since the manuscript is of the review type, I have a few remarks about the References section, since such articles usually receive a lot of attention and are highly cited, everything in this section should be perfect.
Suggestion 1. In some of the references not all authors are indicated (No. 9, 23, 26, 33, 34, 35, 51, 60, 68 and 77), let them all be added.
Responses: Thanks to your suggestion, we have updated the references in the text. We have listed all the authors' names in full. References are numbered 10, 39, 42, 51, 52, 53, 79, 91, 104, 116.
Revised version:
- Berg, G.; Rybakova, D.; Fischer, D.; Cernava, T.; Vergès, M. C.; Charles, T.; Chen, X.; Cocolin, L.; Eversole, K.; Corral, G. H.; Kazou, M.; Kinkel, L.; Lange, L.; Lima, N.; Loy, A.; Macklin, J. A.; Maguin, E.; Mauchline, T.; McClure, R.; Mitter, B.; Ryan, M.; Sarand, I.; Smidt, H.; Schelkle, B.; Roume, H.; Kiran, G. S.; Selvin, J.; Souza, R. S. C.; van Overbeek, L.; Singh, B. K.; Wagner, M.; Walsh, A.; Sessitsch, A.; Schloter, M. Microbiome definition re-visited: old concepts and new challenges. Microbiome 2020, 8, 103, doi:10.1186/s40168-020-00875-0.
- Sato, J.; Kanazawa, A.; Ikeda, F.; Yoshihara, T.; Goto, H.; Abe, H.; Komiya, K.; Kawaguchi, M.; Shimizu, T.; Ogihara, T.; Tamura, Y.; Sakurai, Y.; Yamamoto, R.; Mita, T.; Fujitani, Y.; Fukuda, H.; Nomoto, K.; Takahashi, T.; Asahara, T.; Hirose, T.; Nagata, S.; Yamashiro, Y.; Watada, H. Gut dysbiosis and detection of "live gut bacteria" in blood of Japanese patients with type 2 diabetes. Diabetes care 2014, 37, 2343-2350, doi:10.2337/dc13-2817.
- Pedersen, H. K.; Gudmundsdottir, V.; Nielsen, H. B.; Hyotylainen, T.; Nielsen, T.; Jensen, B. A.; Forslund, K.; Hildebrand, F.; Prifti, E.; Falony, G.; Le Chatelier, E.; Levenez, F.; Doré, J.; Mattila, I.; Plichta, D. R.; Pöhö, P.; Hellgren, L. I.; Arumugam, M.; Sunagawa, S.; Vieira-Silva, S.; Jørgensen, T.; Holm, J. B.; Trošt, K.; Kristiansen, K.; Brix, S.; Raes, J.; Wang, J.; Hansen, T.; Bork, P.; Brunak, S.; Oresic, M.; Ehrlich, S. D.; Pedersen, O. Human gut microbes impact host serum metabolome and insulin sensitivity. Nature 2016, 535, 376-381, doi:10.1038/nature18646.
- Wilson, A. S.; Koller, K. R.; Ramaboli, M. C.; Nesengani, L. T.; Ocvirk, S.; Chen, C. X.; Flanagan, C. A.; Sapp, F. R.; Merritt, Z. T.; Bhatti, F.; Thomas, T. K.; O'Keefe, S. J. D. Diet and the Human Gut Microbiome: An International Review. Digestive Diseases and Sciences 2020, 65, 723-740, doi:10.1007/s10620-020-06112-w.
- Thaiss, C. A.; Levy, M.; Grosheva, I.; Zheng, D.; Soffer, E.; Blacher, E.; Braverman, S.; Tengeler, A. C.; Barak, O.; Elazar, M.; Ben-Zeev, R.; Lehavi-Regev, D.; Katz, M. N.; Pevsner-Fischer, M.; Gertler, A.; Halpern, Z.; Harmelin, A.; Aamar, S.; Serradas, P.; Grosfeld, A.; Shapiro, H.; Geiger, B.; Elinav, E. Hyperglycemia drives intestinal barrier dysfunction and risk for enteric infection. Science (New York, N.Y.) 2018, 359, 1376-1383, doi:10.1126/science.aar3318.
- Mouries, J.; Brescia, P.; Silvestri, A.; Spadoni, I.; Sorribas, M.; Wiest, R.; Mileti, E.; Galbiati, M.; Invernizzi, P.; Adorini, L.; Penna, G.; Rescigno, M. Microbiota-driven gut vascular barrier disruption is a prerequisite for non-alcoholic steatohepatitis development. J Hepatol 2019, 71, 1216-1228, doi:10.1016/j.jhep.2019.08.005.
- Sun, L.; Xie, C.; Wang, G.; Wu, Y.; Wu, Q.; Wang, X.; Liu, J.; Deng, Y.; Xia, J.; Chen, B.; Zhang, S.; Yun, C.; Lian, G.; Zhang, X.; Zhang, H.; Bisson, W. H.; Shi, J.; Gao, X.; Ge, P.; Liu, C.; Krausz, K. W.; Nichols, R. G.; Cai, J.; Rimal, B.; Patterson, A. D.; Wang, X.; Gonzalez, F. J.; Jiang, C. Gut microbiota and intestinal FXR mediate the clinical benefits of metformin. Nature medicine 2018, 24, 1919-1929, doi:10.1038/s41591-018-0222-4.
- Schugar, R. C.; Shih, D. M.; Warrier, M.; Helsley, R. N.; Burrows, A.; Ferguson, D.; Brown, A. L.; Gromovsky, A. D.; Heine, M.; Chatterjee, A.; Li, L.; Li, X. S.; Wang, Z.; Willard, B.; Meng, Y.; Kim, H.; Che, N.; Pan, C.; Lee, R. G.; Crooke, R. M.; Graham, M. J.; Morton, R. E.; Langefeld, C. D.; Das, S. K.; Rudel, L. L.; Zein, N.; McCullough, A. J.; Dasarathy, S.; Tang, W. H. W.; Erokwu, B. O.; Flask, C. A.; Laakso, M.; Civelek, M.; Naga Prasad, S. V.; Heeren, J.; Lusis, A. J.; Hazen, S. L.; Brown, J. M. The TMAO-Producing Enzyme Flavin-Containing Monooxygenase 3 Regulates Obesity and the Beiging of White Adipose Tissue. Cell Rep 2017, 19, 2451-2461, doi:10.1016/j.celrep.2017.05.077.
- Zhang, W. Q.; Zhao, T. T.; Gui, D. K.; Gao, C. L.; Gu, J. L.; Gan, W. J.; Huang, W.; Xu, Y.; Zhou, H.; Chen, W. N.; Liu, Z. L.; Xu, Y. H. Sodium Butyrate Improves Liver Glycogen Metabolism in Type 2 Diabetes Mellitus. J Agric Food Chem 2019, 67, 7694-7705, doi:10.1021/acs.jafc.9b02083.
- de Mello, V. D.; Paananen, J.; Lindström, J.; Lankinen, M. A.; Shi, L.; Kuusisto, J.; Pihlajamäki, J.; Auriola, S.; Lehtonen, M.; Rolandsson, O.; Bergdahl, I. A.; Nordin, E.; Ilanne-Parikka, P.; Keinänen-Kiukaanniemi, S.; Landberg, R.; Eriksson, J. G.; Tuomilehto, J.; Hanhineva, K.; Uusitupa, M. Indolepropionic acid and novel lipid metabolites are associated with a lower risk of type 2 diabetes in the Finnish Diabetes Prevention Study. Scientific reports 2017, 7, 46337, doi:10.1038/srep46337.
Suggestion 2. In reference #18, the names of the authors are abbreviated somewhat too much. Let it be checked and corrected.
Responses: Thank you for your suggestion, we have corrected the reference.
Revised version:
- A, Arora; T, Behl; A, Sehgal; S, Singh; N, Sharma; S, Bhatia; E, Sobarzo-Sanchez; S, Bungau. - Unravelling the involvement of gut microbiota in type 2 diabetes mellitus. D - 0375521 2021, - 119311.
Suggestion 3. My personal opinion is that given that we are in September of 2023, it would be very valuable for the manuscript if there were at least a few references with information from the current year. I'm sure there should be one.
Responses: Thank you for your suggestions, which we have added to the text. Additional reference numbers are 18, 20, 21, 45, 46, 55, 56, 57, 64, 71, 112, 113.
Revised version:
(lines 90-95): Xiao et al. found low levels of phenolic acids and SCFAs in patients with T1D. Phenolic acids have demonstrated antidiabetic and anti-inflammatory effects. The reduced production of SCFAs in T1D patients could be attributed to the decreased abundance of SCFAs-producing flora resulting from the loss of phenolic acids[18]. Therefore, supplementing with phenolic acid compounds may be an effective approach for treating T1D.
- Hu, J.; Ding, J.; Li, X.; Li, J.; Zheng, T.; Xie, L.; Li, C.; Tang, Y.; Guo, K.; Huang, J.; Liu, S.; Yan, J.; Peng, W.; Hou, C.; Wen, L.; Xu, A.; Zhou, Z.; Xiao, Y. Distinct signatures of gut microbiota and metabolites in different types of diabetes: a population-based cross-sectional study. EClinicalMedicine 2023, 62, 102132, doi:10.1016/j.eclinm.2023.102132.
- Frias, J. P.; Lee, M. L.; Carter, M. M.; Ebel, E. R.; Lai, R. H.; Rikse, L.; Washington, M. E.; Sonnenburg, J. L.; Damman, C. J. A microbiome-targeting fibre-enriched nutritional formula is well tolerated and improves quality of life and haemoglobin A1c in type 2 diabetes: A double-blind, randomized, placebo-controlled trial. Diabetes Obes Metab 2023, 25, 1203-1212, doi:10.1111/dom.14967.
- Liu, J.; Zhou, L.; Sun, L.; Ye, X.; Ma, M.; Dou, M.; Shi, L. Association Between Intestinal Prevotella copri Abundance and Glycemic Fluctuation in Patients with Brittle Diabetes. Diabetes, metabolic syndrome and obesity : targets and therapy 2023, 16, 1613-1621, doi:10.2147/dmso.S412872.
(lines 146-152): Farid Najafi et al. conducted a study that highlights the association between Western diets, which include red meat and processed meats, and the pathogenesis of inflammatory diseases, their findings demonstrated that individuals with T2D have significantly higher pro-inflammatory dietary intake than non-T2D, elevated levels of the inflammatory factors IL-1 (Interleukin-1) and TNF-α result from this type of diet, these factors can interfere with insulin signaling, leading to the development of IR[45].
- Namazi, N.; Anjom-Shoae, J.; Najafi, F.; Ayati, M. H.; Darbandi, M.; Pasdar, Y. Pro-inflammatory diet, cardio-metabolic risk factors and risk of type 2 diabetes: A cross-sectional analysis using data from RaNCD cohort study. BMC Cardiovasc Disord 2023, 23, 5, doi:10.1186/s12872-022-03023-8.
(lines 152-156): Demirer et al. conducted a study that investigated the effects of dietary advanced glycation end products (AGEs) on inflammation, their findings suggest that the intake of dietary AGEs can directly or indirectly induce inflammation, given that individuals with diabetes are prone to oxidative stress and inflammation, excessive consumption of dietary AGEs may expedite the inflammatory process[46].
- Demirer, B.; Yardımcı, H.; Erem Basmaz, S. Inflammation level in type 2 diabetes is associated with dietary advanced glycation end products, Mediterranean diet adherence and oxidative balance score: A pathway analysis. J Diabetes Complications 2023, 37, 108354, doi:10.1016/j.jdiacomp.2022.108354.
(lines 227-229): Hu et al. showed that a high-sugar diet induced symptoms of T2D in rats, causing inflammation, disturbances in glucose metabolism and lipid metabolism, as well as a higher abundance of harmful bacteria present in the gut[55].
- Zhao, R.; Li, N.; Liu, W.; Liu, Q.; Zhang, L.; Peng, X.; Zhao, R.; Hu, H. Low glycemic index potato biscuits alleviate physio-histological damage and gut dysbiosis in rats with type-2 diabetes mellitus induced by high-sugar and high-fat diet and streptozotocin. The Journal of nutritional biochemistry 2023, 119, 109401, doi:10.1016/j.jnutbio.2023.109401.
(lines 229-232): Guo et al. found that a Methionine/choline deficient diet resulted in the expression of pro-inflammatory factors in mice, while a Methionine/choline deficient diet resulted in elevated serum insulin levels in mice with symptoms similar to those of T2D[56].
- Zhen, Q.; Liang, Q.; Wang, H.; Zheng, Y.; Lu, Z.; Bian, C.; Zhao, X.; Guo, X. Theabrownin ameliorates liver inflammation, oxidative stress, and fibrosis in MCD diet-fed C57BL/6J mice. Frontiers in endocrinology 2023, 14, 1118925, doi:10.3389/fendo.2023.1118925.
(lines 232-234): Estaphan et al. found that a diet high in iron caused insulin damage and inflammatory cell infiltration in rats, increasing the risk of developing T2D[57].
- Delghingaro-Augusto, V.; Hosaka, A.; Estaphan, S.; Richardson, A.; Dahlstrom, J. E.; Nolan, C. J. High Dietary Iron in Western Diet-Fed Male Rats Causes Pancreatic Islet Injury and Acute Pancreatitis. J Nutr 2023, 153, 723-732, doi:10.1016/j.tjnut.2023.01.009.
(lines 290-292): Moreover, SCFAs has anti-inflammatory properties and can reduce mucosal and chronic organismal inflammation by inhibiting pro-inflammatory cytokines and promoting the secretion of anti-inflammatory cytokines[64].
- McBride, D. A.; Dorn, N. C.; Yao, M.; Johnson, W. T.; Wang, W.; Bottini, N.; Shah, N. J. Short-chain fatty acid-mediated epigenetic modulation of inflammatory T cells in vitro. Drug Deliv Transl Res 2023, 13, 1912-1924, doi:10.1007/s13346-022-01284-6.
(lines 315-316): Furthermore, they influence metabolic inflammation by modulating the expression of cytokines that are pro-inflammatory[71].
- Zhang, Y.; Xi, Y.; Yang, C.; Gong, W.; Wang, C.; Wu, L.; Wang, D. Short-Chain Fatty Acids Attenuate 5-Fluorouracil-Induced THP-1 Cell Inflammation through Inhibiting NF-κB/NLRP3 Signaling via Glycerolphospholipid and Sphingolipid Metabolism. Molecules (Basel, Switzerland) 2023, 28, doi:10.3390/molecules28020494.
(lines 526-529): Increased plasma levels of BCAAs have been associated with higher risks of T2D and insulin resistance in some human examinations[112,113]; therefore, reducing BCAAs intake can restore metabolic health and improve glucose tolerance and insulin sensitivity.
- Supruniuk, E.; Żebrowska, E.; Chabowski, A. Branched chain amino acids-friend or foe in the control of energy substrate turnover and insulin sensitivity? Crit Rev Food Sci Nutr 2023, 63, 2559-2597, doi:10.1080/10408398.2021.1977910.
- Sawicki, K. T.; Ning, H.; Allen, N. B.; Carnethon, M. R.; Wallia, A.; Otvos, J. D.; Ben-Sahra, I.; McNally, E. M.; Snell-Bergeon, J. K.; Wilkins, J. T. Longitudinal trajectories of branched chain amino acids through young adulthood and diabetes in later life. JCI Insight 2023, 8, doi:10.1172/jci.insight.166956.
Thank you again for your suggestions. According to your suggestions, we have made corresponding changes and adjustments in the manuscript. The amendment to the question chapter can be found in the new version.

Round 2
Reviewer 1 Report
Manuscript has substantially improved
no further comments
Author Response
Suggestion 1. Manuscript has substantially improved.
Responses: Thanks for your comments.

Reviewer 2 Report
Dear Authors,
The manuscript has improved by these revisions, but there are still a few comments that should be addressed.
1. Methods (section 7) should be placed in section 2 after introduction.
2. Abbreviations should be after the conclusion and not placed in a table.
3. All tables should be expanded with more details. For example, Effect column in table 1, symptoms of improvement column in table 2, impact on T2D column in table 3.
4. The main important thing is that the manuscript needs extensive editing by an English language speaker.
Extensive English language editing needed.
Author Response
Dear Authors,
The manuscript has improved by these revisions, but there are still a few comments that should be addressed.
Suggestion 1. Methods (section 7) should be placed in section 2 after introduction.
Responses: Thank you for your suggestion, we have put the methods section into section 2 of the article. (lines 109-115)
Revised version:
- Method
As of September 2023, we conducted a literature search in Pubmed and web of science databases, mostly searching for articles published in English in the last 5 years, including reviews and research-based papers. The following keywords were searched: "gut microbial metabolites and T2D", "short-chain fatty acids", "bile acids", "branched-chain amino acids ", "intestinal epithelial barrier dysfunction". To study the effect of gut microbial metabolites on T2D.
Suggestion 2. Abbreviations should be after the conclusion and not placed in a table.
Responses: Thank you for your suggestion, we have put the abbreviation after the conclusion. (lines 587-604)
Revised version:
Abbreviations
T2D, Type 2 diabetes; SCFAs, Short-chain fatty acids; GLP, Glucagon-like peptide; PYY, Peptide YY; TMA, Trimethylamine; TMAO, Trimethylamine N-Oxide; LPS, Lipopolysaccharide; MAM, Microbial anti-inflammatory molecule; BA, Bile acids; IL-6, Interleukin-6; TNF-α, Tumor necrosis factor-α; IL-1β, Interleukin-1β; BCAAs, Branched-chain amino acids; GPCR, G protein-coupled receptor; IL-1, Interleukin-1; AMP, Anti-microbial peptides; IgA, Immunoglobulin A; TJ, Tight junctions; IgG, Immunoglobulin G; GLUT2, Glucose transporter protein; HFD, High-fat diet; PAMPs, Pathogen-associated molecular patterns; IR, Insulin resistance; PPARγ, Peroxisome proliferator-activated receptor γ; iNOS, inducible nitric oxide synthase; FFAR3, G protein-coupled receptors free fatty acid receptor 3; GPR43, Free fatty acid receptor 2; TLRs, Toll-like receptors; NLRP3, NOD-like receptor family pyrin domain containing 3; ZO-1, Zona occludens 1; CA, Cholic acid; CDCA, Chenodeoxycholic acid; CYP7A1, Cholesterol-7α-hydroxylase; CYP27A1, Sterol 27-hydroxylase; TCDCA, Tauro chenodeoxycholic acid; TCA, Taurocholic acid; GCA, Glycocholic acid; GCDCA, Glycochenodeoxycholicacid; BSH, Bile salt hydrolases; DCA, Deoxycholic acid; LCA, Lithocholic acid; UDCA, Ursodeoxycholic acid; FXR, Farnesoid X receptor; VDR, Vitamin D receptor; PXR, Pregnane X receptor; TGR5, G protein-coupled receptor 5; FMO3, Flavin-containing monooxygenase 3.
Suggestion 3. All tables should be expanded with more details. For example, Effect column in table 1, symptoms of improvement column in table 2, impact on T2D column in table 3.
Responses: Thank you for your suggestions, we have added details to all the tables. (line 107, line 511, line 559)
Revised version:
(line 107): Table 1
|
Metabolite |
Microbial agent |
Effect |
Reference |
|
Propionate |
Lactobacillus |
Promotes GLP-1 secretion, improves blood glucose and lipid levels, intestinal barrier function and increases beneficial bacteria. |
[19] |
|
Butyrate |
Anaerostipes hadrus |
Reduces glycated haemoglobin levels, improves mood, sleep and blood sugar levels and increases probiotic abundance. |
[20] |
|
isovaleric, lactic acids |
Prevotella copri |
Improves insulin secretion and promotes glucose homeostasis. |
[21] |
|
BA |
Bacteroides fragilis |
Affects blood sugar levels |
[22] |
|
MAM(Microbial anti-inflammatory molecule) |
Faecalibacterium Prausnitzii |
Regulation of tight junction protein expression, restoration of intestinal barrier function and resistance to inflammation. |
[23] |
|
Acetate |
Akkermansia |
Inhibits intestinal inflammation and promotes intestinal epithelial integrity. |
[24] |
|
SCFA and branched-chain fatty acids |
Acidaminococcus spp, Clostridia spp. |
Promotes healthy metabolism and amino acid fermentation. |
[25][26] |
|
Tryptamine |
Ruminococcus gnavus |
Increases bile acid levels and secretion of tryptamine. |
[27][28] |
|
Histamine |
Morganella morganii |
Decarboxylation of histidine (histidine decarboxylase (HDC)) |
[29][30] |
|
Imidazole propionate (ImP) |
Adlercreutziae equolifaciens Anaerococcus Prevotii |
Impairment of glucose tolerance and insulin signalling. |
[31] |
|
Dopamine |
Enterococcus faecalis |
Regulates glucose uptake, insulin sensitivity and lipid metabolism. |
[32][33] |
(line 511): Table 2
|
Exogenous substances affecting metabolites |
Regulated substance |
Symptoms of improvement |
Reference |
|
Poria oligosaccharides |
BA |
Improves glucose intolerance and IR, lowers blood glucose levels, reduces intestinal damage as well as restores normal microbiota balance. |
[97] |
|
Berberine and Metformin |
LPS |
Reduces food intake, body weight, blood glucose levels, restores SCFA levels, reduces intestinal inflammation as well as restores intestinal barrier function. |
[24] |
|
Legumes Diet |
Gut microorganisms |
Reduces body weight, lowers triglyceride and total cholesterol levels, increases probiotic abundance and regulates glucolipid metabolism. |
[98][99] |
|
Sodium butyrate |
glycogen |
Upregulation of GPR43 and GLUT2 expression promotes hepatocyte glycogen metabolism and maintenance of blood glucose homeostasis. |
[104] |
|
Extract of Sargarsum fusiforme |
BCAAs |
Reducing food and water intake lowers fasting blood glucose levels while improving glucose tolerance and hepatic oxidative stress. |
[105] |
|
Probiotics |
Intestinal proinsulin |
Improves blood glucose and lipid levels, enhances insulin secretion, upregulates claudin-1 and mucin-2 expression, and improves intestinal barrier function. |
[19] |
|
Probiotics and Prebiotics |
Gut microbial metabolites |
Reduces intestinal endotoxin concentrations, energy gain and pro-inflammatory factor levels while increasing insulin sensitivity. |
[100] |
(line 559): Table 3
|
Exogenous substances affecting metabolites |
Regulated metabolites |
Impact on T2D |
Reference |
|
Dietary fibre |
SCFAs |
Improve the growth, survival and metabolic environment of probiotics and promote sugar metabolism. |
[110] |
|
Fruits and grains |
SCFAs |
Reduced food intake, lower levels of inflammation and maintenance of intestinal mucosal homeostasis. |
[111] |
|
Excessive intake of protein |
BCAAs |
Affects glucose tolerance, interferes with metabolism and increases the risk of IR. |
[112][113] |
|
Red Meat Foods |
TMAO |
Increases the risk of T2D, interferes with hepatic insulin signalling pathways, induces adipose tissue inflammation and impairs glucose tolerance. |
[114] |
|
Diet |
Indolepropionic acid |
Maintains normal β cell function, reduces risk of T2D and lowers inflammation levels. |
[116] |
Suggestion 4. The main important thing is that the manuscript needs extensive editing by an English language speaker.
Responses: Thank you for your suggestions, we have made improvements to the article.
